# Efficient Dictionary Learning with Gradient Descent

## Abstract

Randomly initialized first-order optimization algorithms are the method of choice for solving many high-dimensional nonconvex problems in machine learning, yet general theoretical guarantees cannot rule out convergence to critical points of poor objective value. For some highly structured nonconvex problems however, the success of gradient descent can be understood by studying the geometry of the objective. We study one such problem – complete orthogonal dictionary learning, and provide converge guarantees for randomly initialized gradient descent to the neighborhood of a global optimum. The resulting rates scale as low order polynomials in the dimension even though the objective possesses an exponential number of saddle points. This efficient convergence can be viewed as a consequence of negative curvature normal to the stable manifolds associated with saddle points, and we provide evidence that this feature is shared by other nonconvex problems of importance as well.

## 1 Introduction

Many central problems in machine learning and signal processing are most naturally formulated as optimization problems. These problems are often both nonconvex and high-dimensional. High dimensionality makes the evaluation of second-order information prohibitively expensive, and thus randomly initialized first-order methods are usually employed instead. This has prompted great interest in recent years in understanding the behavior of gradient descent on nonconvex objectives (18; 14; 17; 11). General analysis of first- and second-order methods on such problems can provide guarantees for convergence to critical points but these may be highly suboptimal, since nonconvex optimization is in general an NP-hard probem (4). Outside of a convex setting (28) one must assume additional structure in order to make statements about convergence to optimal or high quality solutions. It is a curious fact that for certain classes of problems such as ones that involve sparsification (25; 6) or matrix/tensor recovery (21; 19; 1) first-order methods can be used effectively. Even for some highly nonconvex problems where there is no ground truth available such as the training of neural networks first-order methods converge to high-quality solutions (40).

Dictionary learning is a problem of inferring a sparse representation of data that was originally developed in the neuroscience literature (30), and has since seen a number of important applications including image denoising, compressive signal acquisition and signal classification (13; 26). In this work we study a formulation of the dictionary learning problem that can be solved efficiently using randomly initialized gradient descent despite possessing a number of saddle points exponential in the dimension. A feature that appears to enable efficient optimization is the existence of sufficient negative curvature in the directions normal to the stable manifolds of all critical points that are not global minima [1]. This property ensures that the regions of the space that feed into small gradient regions under gradient flow do not dominate the parameter space. Figure 1 illustrates the value of this property: negative curvature prevents measure from concentrating about the stable manifold. As a consequence randomly initialized gradient methods avoid the "slow region" of around the saddle point.

---

[1] As well as a lack of spurious local minimizers, and the existence of large gradients or strong convexity in the remaining parts of the space

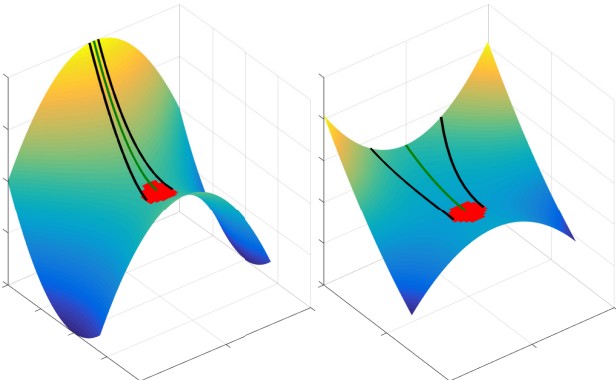

Figure 1: **Negative curvature helps gradient descent.** Red: "slow region" of small gradient around a saddle point. Green: stable manifold associated with the saddle point. Black: points that flow to the slow region. Left: global negative curvature normal to the stable manifold. Right: positive curvature normal to the stable manifold – randomly initialized gradient descent is more likely to encounter the slow region.

The main results of this work is a convergence rate for randomly initialized gradient descent for complete orthogonal dictionary learning to the neighborhood of a global minimum of the objective. Our results are probabilistic since they rely on initialization in certain regions of the parameter space, yet they allow one to flexibly trade off between the maximal number of iterations in the bound and the probability of the bound holding.

While our focus is on dictionary learning, it has been recently shown that for other important nonconvex problems such as phase retrieval (8) performance guarantees for randomly initialized gradient descent can be obtained as well. In fact, in Appendix C we show that negative curvature normal to the stable manifolds of saddle points (illustrated in Figure 1) is also a feature of the population objective of generalized phase retrieval, and can be used to obtain an efficient convergence rate.

## 2 RELATED WORK

**Easy nonconvex problems.** There are two basic impediments to solving nonconvex problems globally: **(i) spurious local minimizers**, and **(ii) flat saddle points**, which can cause methods to stagnate in the vicinity of critical points that are not minimizers. The latter difficulty has motivated the study of *strict saddle functions* (36; 14), which have the property that at every point in the domain of optimization, there is a large gradient, a direction of strict negative curvature, or the function is strongly convex. By leveraging this curvature information, it is possible to escape saddle points and obtain a local minimizer in polynomial time.[2] Perhaps more surprisingly, many known strict saddle functions also have the property that every local minimizer is global; for these problems, this implies that efficient methods find global solutions. Examples of problems with this property include variants of sparse dictionary learning (38), phase retrieval (37), tensor decomposition (14), community detection (3) and phase synchronization (5).

**Minimizing strict saddle functions.** Strict saddle functions have the property that at every saddle point there is a direction of strict negative curvature. A natural approach to escape such saddle points is to use second order methods (e.g., trust region (9) or curvilinear search (15)) that explicitly leverage curvature information. Alternatively, one can attempt to escape saddle points using first order information only. However, some care is needed: canonical first order methods such as gradient descent will not obtain minimizers if initialized at a saddle point (or at a point that flows to one) – at any critical point, gradient descent simply stops. A natural remedy is to randomly perturb the iterate whenever needed. A line of recent works shows that noisy gradient methods of this form efficiently optimize strict saddle functions (24; 12; 20). For example, (20) obtains rates on strict saddle functions that match the optimal rates for smooth convex programs up to a polylogarithmic dependence on dimension.[3]

---

[2]This statement is nontrivial: finding a local minimum of a smooth function is NP-hard.

[3]This work also proves convergence to a second-order stationary point under more general smoothness assumptions.

**Randomly initialized gradient descent?** The aforementioned results are broad, and nearly optimal. Nevertheless, important questions about the behavior of first order methods for nonconvex optimization remain unanswered. For example: *in every one of the aforemented benign nonconvex optimization problems, randomly initialized gradient descent rapidly obtains a minimizer.* This may seem unsurprising: general considerations indicate that the stable manifolds associated with non-minimizing critical points have measure zero (29), this implies that a variety of small-stepping first order methods converge to minimizers in the large-time limit (23). However, it is not difficult to construct strict saddle problems that *are not* amenable to efficient optimization by randomly initialized gradient descent – see (12) for an example. This contrast between the excellent empirical performance of randomly initialized first order methods and worst case examples suggests that there are important geometric and/or topological properties of "easy nonconvex problems" that are not captured by the strict saddle hypothesis. Hence, the motivation of this paper is twofold: (i) to provide theoretical corroboration (in certain specific situations) for what is arguably the simplest, most natural, and most widely used first order method, and (ii) to contribute to the ongoing effort to identify conditions which make nonconvex problems amenable to efficient optimization.

## 3 Dictionary Learning over the Sphere

Suppose we are given data matrix $\boldsymbol{Y} = \begin{bmatrix} \boldsymbol{y}_1, \ldots \boldsymbol{y}_p \end{bmatrix} \in \mathbb{R}^{n \times p}$. The *dictionary learning* problem asks us to find a concise representation of the data (13), of the form $\boldsymbol{Y} \approx \boldsymbol{AX}$, where $\boldsymbol{X}$ is a sparse matrix. In the *complete, orthogonal dictionary learning* problem, we restrict the matrix $\boldsymbol{A}$ to have orthonormal columns ($\boldsymbol{A} \in O(n)$). This variation of dictionary learning is useful for finding concise representations of small datasets (e.g., patches from a single image, in MRI (32)).

To analyze the behavior of dictionary learning algorithms theoretically, it useful to posit that $\boldsymbol{Y} = \boldsymbol{A}_0 \boldsymbol{X}_0$ for some true dictionary $\boldsymbol{A}_0 \in O(n)$ and sparse coefficient matrix $\boldsymbol{X}_0 \in \mathbb{R}^{n \times p}$, and ask whether a given algorithm recovers the pair $(\boldsymbol{A}_0, \boldsymbol{X}_0)$.[4] In this work, we further assume that the sparse matrix $\boldsymbol{X}_0$ is random, with entries i.i.d. Bernoulli-Gaussian[5]. For simplicity, we will let $\boldsymbol{A}_0 = \boldsymbol{I}$; our arguments extend directly to general $\boldsymbol{A}_0$ via the simple change of variables $\boldsymbol{q} \mapsto \boldsymbol{A}_0^* \boldsymbol{q}$.

(34) showed that under mild conditions, the complete dictionary recovery problem can be reduced to the geometric problem of finding a sparse vector in a linear subspace (31). Notice that because $\boldsymbol{A}_0$ is orthogonal, $\text{row}(\boldsymbol{Y}) = \text{row}(\boldsymbol{X}_0)$. Because $\boldsymbol{X}_0$ is a sparse random matrix, the rows of $\boldsymbol{X}_0$ are sparse vectors. Under mild conditions (34), they are the *sparsest* vectors in the row space of $\boldsymbol{Y}$, and hence can be recovered by solving the conceptual optimization problem

$$\min \ \|\boldsymbol{q}^* \boldsymbol{Y}\|_0 \quad \text{s.t.} \quad \boldsymbol{q}^* \boldsymbol{Y} \neq \boldsymbol{0}.$$

This is not a well-structured optimization problem: the objective is discontinuous, and the constraint set is open. A natural remedy is to replace the $\ell^0$ norm with a smooth sparsity surrogate, and to break the scale ambiguity by constraining $\boldsymbol{q}$ to the sphere, giving

$$\min \ f_{\text{DL}}(\boldsymbol{q}) \equiv \frac{1}{p} \sum_{k=1}^{p} h_\mu(\boldsymbol{q}^* \boldsymbol{y}_k) \quad \text{s.t.} \quad \boldsymbol{q} \in \mathbb{S}^{n-1}. \tag{1}$$

Here, we choose $h_\mu(t) = \mu \log(\cosh(t/\mu))$ as a smooth sparsity surrogate. This objective was analyzed in (35), which showed that (i) although this optimization problem is nonconvex, when the data are sufficiently large, with high probability every local optimizer is near a signed column of the true dictionary $\boldsymbol{A}_0$, (ii) every other critical point has a direction of strict negative curvature, and (iii) as a consequence, a second-order Riemannian trust region method efficiently recovers a column of $\boldsymbol{A}_0$.[6] The Riemannian trust region method is of mostly theoretical interest: it solves complicated (albeit polynomial time) subproblems that involve the Hessian of $f_{\text{DL}}$.

---

[4]This problem exhibits a sign permutation symmetry: $\boldsymbol{A}_0 \boldsymbol{X}_0 = (\boldsymbol{A}_0 \boldsymbol{\Gamma})(\boldsymbol{\Gamma}^* \boldsymbol{X}_0)$ for any signed permutation matrix $\boldsymbol{\Gamma}$. Hence, we only ask for recovery up to a signed permutation.

[5]$[\boldsymbol{X}_0]_{ij} = \boldsymbol{V}_{ij} \boldsymbol{\Omega}_{ij}$, with $\boldsymbol{V}_{ij} \sim \mathcal{N}(0,1)$, $\boldsymbol{\Omega}_{ij} \sim \text{Bern}(\theta)$ independent.

[6]Combining with a deflation strategy, one can then efficiently recover the entire dictionary $\boldsymbol{A}_0$.

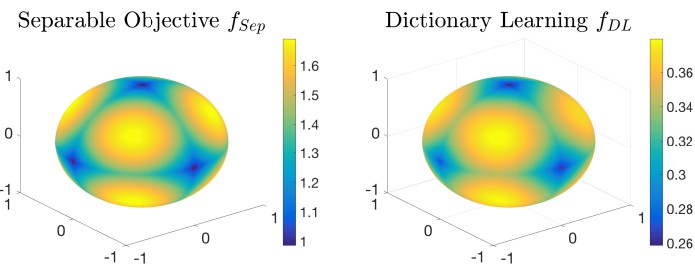

Figure 2: *Left:* The separable objective for $n = 3$. Note the similarity to the dictionary learning objective. *Right:* The objective for complete orthogonal dictionary learning (discussed in section 6) for $n = 3$.

In practice, simple iterative methods, including randomly initialized gradient descent are also observed to rapidly obtain high-quality solutions. In the sequel, we will give a geometric explanation for this phenomenon, and bound the rate of convergence of randomly initialized gradient descent to the neighborhood of a column of $\boldsymbol{A}_0$. Our analysis of $f_{\mathrm{DL}}$ is probabilistic in nature: it argues that with high probability in the sparse matrix $\boldsymbol{X}_0$, randomly initialized gradient descent rapidly produces a minimizer.

To isolate more clearly the key intuitions behind this analysis, we first analyze the simpler *separable objective*

$$\min \; f_{\mathrm{Sep}}(\boldsymbol{q}) \equiv \sum_{i=1}^{n} h_\mu(\boldsymbol{q}_i) \quad \text{s.t.} \quad \boldsymbol{q} \in \mathbb{S}^{n-1}. \tag{2}$$

Figure 2 plots both $f_{\mathrm{Sep}}$ and $f_{\mathrm{DL}}$ as functions over the sphere. Notice that many of the key geometric features in $f_{\mathrm{DL}}$ are present in $f_{\mathrm{Sep}}$; indeed, $f_{\mathrm{Sep}}$ can be seen as an "ultrasparse" version of $f_{\mathrm{DL}}$ in which the columns of the true sparse matrix $\boldsymbol{X}_0$ are taken to have only one nonzero entry. A virtue of this model function is that its critical points and their stable manifolds have simple closed form expressions (see Lemma 1).

## 4 Outline of Important Geometric Features

Our problems of interest have the form

$$\min f(\boldsymbol{q}) \quad \text{s.t.} \quad \boldsymbol{q} \in \mathbb{S}^{n-1},$$

where $f : \mathbb{R}^n \to \mathbb{R}$ is a smooth function. We let $\nabla f(\boldsymbol{q})$ and $\nabla^2 f(\boldsymbol{q})$ denote the Euclidean gradient and hessian (over $\mathbb{R}^n$), and let $\mathrm{grad}\,[f]\,(\boldsymbol{q})$ and $\mathrm{Hess}\,[f]\,(\boldsymbol{q})$ denote their Riemannian counterparts (over $\mathbb{S}^{n-1}$). We will obtain results for Riemannian gradient descent defined by the update

$$\boldsymbol{q} \to \exp_{\boldsymbol{q}}(-\eta\,\mathrm{grad}[f](\boldsymbol{q}))$$

for some step size $\eta > 0$, where $\exp_{\boldsymbol{q}} : T_{\boldsymbol{q}}\mathbb{S}^{n-1} \to \mathbb{S}^{n-1}$ is the exponential map. The Riemannian gradient on the sphere is given by $\mathrm{grad}[f](\boldsymbol{q}) = (\boldsymbol{I} - \boldsymbol{q}\boldsymbol{q}^*)\nabla f(\boldsymbol{q})$.

We let $A$ denote the set of critical points of $f$ over $\mathbb{S}^{n-1}$ – these are the points $\bar{\boldsymbol{q}}$ s.t. $\mathrm{grad}\,[f]\,(\bar{\boldsymbol{q}}) = \boldsymbol{0}$. We let $\breve{A}$ denote the set of local minimizers, and $\hat{A}$ its complement. Both $f_{\mathrm{Sep}}$ and $f_{\mathrm{DL}}$ are *Morse functions* on $\mathbb{S}^{n-1}$,[7] we can assign an index $\alpha$ to every $\bar{\boldsymbol{q}} \in A$, which is the number of negative eigenvalues of $\mathrm{Hess}\,[f]\,(\bar{\boldsymbol{q}})$.

Our goal is to understand when gradient descent efficiently converges to a local minimizer. In the small-step limit, gradient descent follows gradient flow lines $\boldsymbol{\gamma} : \mathbb{R} \to \mathcal{M}$, which are solution curves of the ordinary differential equation

$$\dot{\boldsymbol{\gamma}}(t) = -\mathrm{grad}\,[f]\,(\boldsymbol{\gamma}(t))$$

To each critical point $\boldsymbol{\alpha} \in A$ of index $\lambda$, there is an associated *stable manifold* of dimension $\dim(\mathcal{M}) - \lambda$, which is roughly speaking, the set of points that flow to $\alpha$ under gradient flow:

$$W^s(\boldsymbol{\alpha}) \equiv \left\{ \boldsymbol{q} \in \mathcal{M} \;\middle|\; \begin{array}{l} \lim_{t\to\infty} \boldsymbol{\gamma}(t) = \boldsymbol{\alpha} \\ \boldsymbol{\gamma} \text{ a gradient flow line s.t. } \boldsymbol{\gamma}(0) = \boldsymbol{q} \end{array} \right\}.$$

---

[7]Strictly speaking, $f_{\mathrm{DL}}$ is Morse with high probability, due to results of (38).

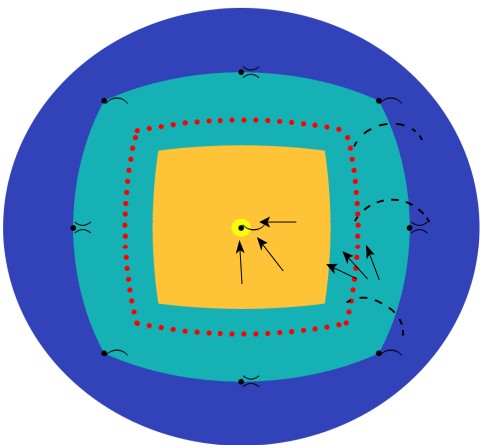

Figure 3: **Negative curvature and efficient gradient descent.** The union of the light blue, orange and yellow sets is the set $\mathcal{C}$. In the light blue region, there is negative curvature normal to $\partial\mathcal{C}$, while in the orange region the gradient norm is large, as illustrated by the arrows. There is a single global minimizer in the yellow region. For the separable objective, the stable manifolds of the saddles and maximizers all lie on $\partial\mathcal{C}$ (the black circles denote the critical points, which are either maximizers "$\frown$", saddles "$\asymp$", or minimizers "$\smile$"). The red dots denote $\partial\mathcal{C}_\zeta$ with $\zeta = 0.2$.

Our analysis uses the following convenient coordinate chart

$$\boldsymbol{\varphi}(\boldsymbol{w}) = \left(\boldsymbol{w}, \sqrt{1 - \|\boldsymbol{w}\|^2}\right) \equiv \boldsymbol{q}(\boldsymbol{w}) \tag{3}$$

where $\boldsymbol{w} \in B_1(0)$. We also define two useful sets:

$$\mathcal{C} \equiv \{\boldsymbol{q} \in \mathbb{S}^{n-1} | q_n \geq \|\boldsymbol{w}\|_\infty\}$$

$$\mathcal{C}_\zeta \equiv \left\{\boldsymbol{q} \in \mathbb{S}^{n-1} \,\middle|\, \frac{q_n}{\|\boldsymbol{w}\|_\infty} \geq 1 + \zeta\right\}. \tag{4}$$

Since the problems considered here are symmetric with respect to a signed permutation of the coordinates we can consider a certain $\mathcal{C}$ and the results will hold for the other symmetric sections as well. We will show that at every point in $\mathcal{C}$ aside from a neighborhood of a global minimizer for the separable objective (or a solution to the dictionary problem that may only be a local minimizer), there is either a large gradient component in the direction of the minimizer or negative curvature in a direction normal to $\partial\mathcal{C}$. For the case of the separable objective, one can show that the stable manifolds of the saddles lie on this boundary, and hence this curvature is normal to the stable manifolds of the saddles and allows rapid progress away from small gradient regions and towards a global minimizer [8]. These regions are depicted in Figure 3.

In the sequel, we will make the above ideas precise for the two specific nonconvex optimization problems discussed in Section 3 and use this to obtain a convergence rate to a neighborhood of a global minimizer. Our analysis are specific to these problems. However, as we will describe in more detail later, they hinge on important geometric characteristics of these problems which make them amenable to efficient optimization, which may obtain in much broader classes of problems.

---

[8]The direction of this negative curvature is important here, and it is this feature that distinguishes these problems from other problems in the strict-saddle class where this direction may be arbitrary

## 5 SEPARABLE FUNCTION CONVERGENCE RATE

In this section, we study the behavior of randomly initialized gradient descent on the separable function $f_{\text{Sep}}$. We begin by characterizing the critical points:

**Lemma 1** (Critical points of $f_{\text{Sep}}$). *The critical points of the separable problem* (2) *are*

$$A = \left\{ \mathcal{P}_{\mathbb{S}^{n-1}}[\boldsymbol{a}] \,\middle|\, \boldsymbol{a} \in \{-1, 0, 1\}^{\otimes n}, \|\boldsymbol{a}\| > 0 \right\}. \tag{5}$$

*For every* $\boldsymbol{\alpha} \in A$ *and corresponding* $\boldsymbol{a}(\boldsymbol{\alpha})$, *for* $\mu < \frac{c}{\sqrt{n}\log n}$ *the stable manifold of* $\boldsymbol{\alpha}$ *takes the form*

$$W^s(\boldsymbol{\alpha}) = \left\{ \mathcal{P}_{\mathbb{S}^{n-1}}\left[\boldsymbol{a}(\boldsymbol{\alpha}) + \boldsymbol{b}\right] \,\middle|\, \begin{array}{c} \text{supp}(\boldsymbol{a}(\boldsymbol{\alpha})) \cap \text{supp}(\boldsymbol{b}) = \varnothing, \\ \|\boldsymbol{b}\|_\infty < 1 \end{array} \right\} \tag{6}$$

*where* $c > 0$ *is a numerical constant.*

*Proof.* Please see Appendix A $\qquad\qquad\square$

By inspecting the dimension of the stable manifolds, it is easy to verify that that there are $2n$ global minimizers at the 1-sparse vectors on the sphere $\pm\widehat{\boldsymbol{e}}_i$, $2^n$ maximizers at the least sparse vectors and an exponential number of saddle points of intermediate sparsity. This is because the dimension of $W^s(\alpha)$ is simply the dimension of $b$ in 6, and it follows directly from the stable manifold theorem that only minimizers will have a stable manifold of dimension $n - 1$. The objective thus possesses no spurious local minimizers.

When referring to critical points and stable manifolds from now on we refer only to those that are contained in $\mathcal{C}$ or on its boundary. It is evident from Lemma 1 that the critical points in $\hat{A}$ all lie on $\partial\mathcal{C}$ and that $\bigcup_{\boldsymbol{\alpha}\in\hat{A}} W^s(\boldsymbol{\alpha}) = \partial\mathcal{C}$ , and there is a minimizer at its center given by $\boldsymbol{q}(\boldsymbol{0}) = \widehat{\boldsymbol{e}}_n$.

### 5.1 THE EFFECT OF NEGATIVE CURVATURE ON THE GRADIENT

We now turn to making precise the notion that negative curvature normal to stable manifolds of saddle points enables gradient descent to rapidly exit small gradient regions. We do this by defining vector fields $\boldsymbol{u}^{(i)}(\boldsymbol{q}), i \in [n-1]$ such that each field is normal to a continuous piece of $\partial\mathcal{C}_\zeta$ and points outwards relative to $\mathcal{C}_\zeta$ defined in 4. By showing that the Riemannian gradient projected in this direction is positive and proportional to $\zeta$, we are then able to show that gradient descent acts to increase $\zeta(\boldsymbol{q}(\boldsymbol{w})) = \frac{q_n}{\|\boldsymbol{w}\|_\infty} - 1$ geometrically. This corresponds to the behavior illustrated in the light blue region in Figure 3.

**Lemma 2** (Separable objective gradient projection). *For any* $\boldsymbol{w} \in \mathcal{C}_\zeta, i \in [n-1]$, *we define a vector* $\boldsymbol{u}^{(i)} \in T_{\boldsymbol{q}(\boldsymbol{w})}\mathbb{S}^{n-1}$ *by*

$$u_j^{(i)} = \begin{cases} 0 & j \notin \{i, n\}, \\ \text{sign}(w_i) & j = i, \\ -\frac{|w_i|}{q_n} & j = n. \end{cases} \tag{7}$$

*If* $\mu\log\left(\frac{1}{\mu}\right) \le w_i$ *and* $\mu < \frac{1}{16}$, *then*

$$\boldsymbol{u}^{(i)*}\text{grad}[f_{\text{Sep}}](\boldsymbol{q}(\boldsymbol{w})) \ge c\|\boldsymbol{w}\|_\infty \zeta,$$

*where* $c > 0$ *is a numerical constant.*

*Proof.* Please see Appendix A. $\qquad\qquad\square$

Since we will use this property of the gradient in $\mathcal{C}_\zeta$ to derive a convergence rate, we will be interested in bounding the probability that gradient descent initialized randomly with respect to a uniform measure on the sphere is initialized in $\mathcal{C}_\zeta$. This will require bounding the volume of this set, which is done in the following lemma:

**Lemma 3** (Volume of $\mathcal{C}_\zeta$). *For $\mathcal{C}_\zeta$ defined as in (4) we have*

$$\frac{\mathrm{Vol}(\mathcal{C}_\zeta)}{\mathrm{Vol}(\mathbb{S}^{n-1})} \geq \frac{1}{2n} - \frac{\log(n)}{n}\zeta$$

*Proof.* Please see Appendix D.3. □

### 5.2 Convergence rate

Using the results above, one can obtain the following convergence rate:

**Theorem 1** (Gradient descent convergence rate for separable function). *For any $0 < \zeta_0 < 1$, $r > \mu \log\left(\frac{1}{\mu}\right)$, Riemannian gradient descent with step size $\eta < \min\left\{\frac{c_1}{n}, \frac{\mu}{2}\right\}$ on the separable objective (2) with $\mu < \frac{c_2}{\sqrt{n}\log n}$, enters an $L^\infty$ ball of radius $r$ around a global minimizer in*

$$T < \frac{C}{\eta}\left(\frac{\sqrt{n}}{r^2} + \log\left(\frac{1}{\zeta_0}\right)\right)$$

*iterations with probability*

$$\mathbb{P} \geq 1 - 2\log(n)\zeta_0,$$

*where $c_i, C > 0$ are numerical constants.*

*Proof.* Please see Appendix A. □

We have thus obtained a convergence rate for gradient descent that relies on the negative curvature around the stable manifolds of the saddles to rapidly move from these regions of the space towards the vicinity of a global minimizer. This is evinced by the logarithmic dependence of the rate on $\zeta$. As was shown for orthogonal dictionary learning in (38), we also expect a linear convergence rate due to strong convexity in the neighborhood of a minimizer, but do not take this into account in the current analysis.

## 6 Dictionary Learning Convergence Rate

The proofs in this section will be along the same lines as those of Section 5. While we will not describe the positions of the critical points explicitly, the similarity between this objective and the separable function motivates a similar argument. It will be shown that initialization in some $\mathcal{C}_\zeta$ will guarantee that Riemannian gradient descent makes uniform progress in function value until reaching the neighborhood of a global minimizer. We will first consider the population objective which corresponds to the infinite data limit

$$f_{\mathrm{DL}}^{\mathrm{pop}}(\boldsymbol{q}) \equiv \mathop{\mathbb{E}}_{\boldsymbol{X}_0} f_{\mathrm{DL}}(\boldsymbol{q}) = \mathbb{E}_{\boldsymbol{x}\sim\mathrm{i.i.d.\,BG}(\theta)}\left[h_\mu(\boldsymbol{q}^*\boldsymbol{x})\right]. \tag{8}$$

and then bounding the finite sample size fluctuations of the relevant quantities. We begin with a lemma analogous to Lemma 2:

**Lemma 4** (Dictionary learning population gradient). *For $\boldsymbol{w} \in \mathcal{C}_\zeta, r < |w_i|, \mu < c_1 r^{5/2}\sqrt{\zeta}$ the dictionary learning population objective 8 obeys*

$$\boldsymbol{u}^{(i)*}\mathrm{grad}[f_{\mathrm{DL}}^{\mathrm{pop}}](\boldsymbol{q}(\boldsymbol{w})) \geq c_\theta r^3 \zeta$$

*where $c_\theta$ depends only on $\theta$, $c_1$ is a positive numerical constant and $\boldsymbol{u}^{(i)}$ is defined in 7.*

*Proof.* Please see Appendix B □

Using this result, we obtain the desired convergence rate for the population objective, presented in Lemma 11 in Appendix B. After accounting for finite sample size fluctuations in the gradient, one obtains a rate of convergence to the neighborhood of a solution (which is some signed basis vector due to our choice $\boldsymbol{A}_0 = \boldsymbol{I}$)

**Theorem 2** (Gradient descent convergence rate for dictionary learning)**.** *For any $1 > \zeta_0 > 0, s > \frac{\mu}{4\sqrt{2}}$, Riemannian gradient descent with step size $\eta < \frac{c_5 \theta s}{n \log np}$ on the dictionary learning objective 1 with $\mu < \frac{c_6 \sqrt{\zeta_0}}{n^{5/4}}, \theta \in (0, \frac{1}{2})$, enters a ball of radius $c_3 s$ from a target solution in*

$$T < \frac{C_2}{\eta\theta} \left( \frac{1}{s} + n \log \frac{1}{\zeta_0} \right)$$

*iterations with probability*

$$\mathbb{P} \geq 1 - 2\log(n)\zeta_0 - \mathbb{P}_y - c_8 p^{-6}$$

*where $y = \frac{c_7 \theta(1-\theta)\zeta_0}{n^{3/2}}$, $\mathbb{P}_y$ is given in Lemma 10 and $c_i, C_i$ are positive constants.*

*Proof.* Please see Appendix B ◻

The two terms in the rate correspond to an initial geometric increase in the distance from the set containing the small gradient regions around saddle points, followed by convergence to the vicinity of a minimizer in a region where the gradient norm is large. The latter is based on results on the geometry of this objective provided in (38).

## 7    Discussion

The above analysis suggests that second-order properties - namely negative curvature normal to the stable manifolds of saddle points - play an important role in the success of randomly initialized gradient descent in the solution of complete orthogonal dictionary learning. This was done by furnishing a convergence rate guarantee that holds when the random initialization is not in regions that feed into small gradient regions around saddle points, and bounding the probability of such an initialization. In Appendix C we provide an additional example of a nonconvex problem that for which an efficient rate can be obtained based on an analysis that relies on negative curvature normal to stable manifolds of saddles - generalized phase retrieval. An interesting direction of further work is to more precisely characterize the class of functions that share this feature.

The effect of curvature can be seen in the dependence of the maximal number of iterations $T$ on the parameter $\zeta_0$. This parameter controlled the volume of regions where initialization would lead to slow progress and the failure probability of the bound $1 - \mathbb{P}$ was linear in $\zeta_0$, while $T$ depended logarithmically on $\zeta_0$. This logarithmic dependence is due to a geometric increase in the distance from the stable manifolds of the saddles during gradient descent, which is a consequence of negative curvature. Note that the choice of $\zeta_0$ allows one to flexibly trade off between $T$ and $1 - \mathbb{P}$. By decreasing $\zeta_0$, the bound holds with higher probability, at the price of an increase in $T$. This is because the volume of acceptable initializations now contains regions of smaller minimal gradient norm. In a sense, the result is an extrapolation of works such as (23) that analyze the $\zeta_0 = 0$ case to finite $\zeta_0$.

Our analysis uses precise knowledge of the location of the stable manifolds of saddle points. For less symmetric problems, including variants of sparse blind deconvolution (41) and overcomplete tensor decomposition, there is no closed form expression for the stable manifolds. However, it is still possible to coarsely localize them in regions containing negative curvature. Understanding the implications of this geometric structure for randomly initialized first-order methods is an important direction for future work.

One may hope that studying simple model problems and identifying structures (here, negative curvature orthogonal to the stable manifold) that enable efficient optimization will inspire approaches to broader classes of problems. One problem of obvious interest is the training of deep neural networks for classification, which shares certain high-level features with the problems discussed in this paper. The objective is also highly nonconvex and is conjectured to contain a proliferation of saddle points (11), yet these appear to be avoided by first-order methods (16) for reasons that are still quite poorly understood beyond the two-layer case (39).

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

## A   Proofs - Separable Objective

**Proof of Lemma 1: (Critical point structure of separable objective)**. Denoting by $\tanh(\frac{\boldsymbol{q}}{\mu})$ a vector in $\mathbb{R}^n$ elements $\tanh(\frac{\boldsymbol{q}}{\mu})_i = \tanh(\frac{q_i}{\mu})$ we have

$$\text{grad}[f_{Sep}](\boldsymbol{q})_i = (\boldsymbol{I} - \boldsymbol{q}\boldsymbol{q}^*)\tanh(\frac{\boldsymbol{q}}{\mu})$$

. Thus critical points are ones where either $\tanh(\frac{\boldsymbol{q}}{\mu}) = \boldsymbol{0}$ (which cannot happen on $\mathbb{S}^{n-1}$) or $\tanh(\frac{\boldsymbol{q}}{\mu})$ is in the nullspace of $(\boldsymbol{I} - \boldsymbol{q}\boldsymbol{q}^*)$, which implies $\tanh(\frac{\boldsymbol{q}}{\mu}) = c\boldsymbol{q}$ for some constant $b$. The equation $\tanh(\frac{x}{\mu}) = bx$ has either a single solution at the origin or 3 solutions at $\{0, \pm r(b)\}$ for some $r(b)$. Since this equation must be solves simultaneously for every element of $\boldsymbol{q}$, we obtain $\forall i \in [n] : q_i \in \{0, \pm r(b)\}$. To obtain solutions on the sphere, one then uses the freedom we have in choosing $b$ (and thus $r(b)$) such that $\|\boldsymbol{q}\| = 1$. The resulting set of critical points is thus

$$A = \mathcal{P}_{\mathbb{S}^{n-1}}\Big[\ \{-1, 0, 1\}^n \setminus \{\boldsymbol{0}\}\ \Big].$$

To prove the form of the stable manifolds, we first show that for $q_i$ such that $|q_i| = \|\boldsymbol{q}\|_\infty$ and any $q_j$ such that $|q_j| + \Delta = |q_i|$ and sufficiently small $\Delta > 0$, we have

$$-\text{grad}[f_{Sep}](\boldsymbol{q})_i\text{sign}(q_i) > -\text{grad}[f_{Sep}](\boldsymbol{q})_i\text{sign}(q_j) \tag{9}$$

For ease of notation we now assume $q_i, q_j > 0$ and hence $\Delta = q_i - q_j$, otherwise the argument can be repeated exactly with absolute values instead. The above inequality can then be written as

$$\underbrace{(q_i - q_j)\sum_{k=1}^{n}\tanh(\frac{q_k}{\mu})q_k - \tanh(\frac{q_i}{\mu}) + \tanh(\frac{q_j}{\mu})}_{\equiv h} > 0.$$

If we now define $s^2 = \sum_{\substack{k=1 \\ k \neq i, n}}^{n-1} q_k^2$ and $q_n = \sqrt{1 - s^2 - (q_j + \Delta)^2}$ we have

$$h = \Delta\left(\begin{array}{c}\tanh(\frac{q_j+\Delta}{\mu})\,(q_j + \Delta) + \\ \tanh(\frac{\sqrt{1-s^2-(q_j+\Delta)^2}}{\mu})\sqrt{1 - s^2 - (q_j + \Delta)^2} \\ +\Delta\sum_{k\neq i,n}\tanh(\frac{q_k}{\mu})q_k - \tanh(\frac{q_j+\Delta}{\mu}) + \tanh(\frac{q_j}{\mu})\end{array}\right)$$

$$= \Delta\left(\underbrace{\begin{array}{c}\sum_{k\neq i,n}\tanh(\frac{q_k}{\mu})q_k + \tanh(\frac{q_j}{\mu})q_j \\ + \tanh(\frac{\sqrt{1-s^2-q_j^2}}{\mu})\sqrt{1 - s^2 - q_j^2}\end{array}}_{\equiv h_1} \underbrace{- \text{sech}^2(\frac{q_j}{\mu})\frac{1}{\mu}}_{\equiv h_2}\right) + O(\Delta^2)$$

where the $O(\Delta^2)$ term is bounded. Defining a vector $\boldsymbol{r} \in \mathbb{R}^n$ by

$$k \neq i, n : r_k = q_k, r_i = \tanh(\frac{q_j}{\mu})q_j, r_n = \sqrt{1 - s^2 - q_j^2}$$

we have $\|\boldsymbol{r}\|^2 = 1$. Since $\tanh(x)$ is concave for $x > 0$, and $|r_i| \leq 1$, we find

$$h_1 = \sum_{k=1}^{n} \tanh(\frac{r_k}{\mu}) r_k \geq \tanh(\frac{1}{\mu}) \sum_{k=1}^{n} r_k^2 = \tanh(\frac{1}{\mu}).$$

From $|q_i| = \|\boldsymbol{q}\|_\infty$ it follows that $q_i \geq \frac{1}{\sqrt{n}}$ and thus $q_j \geq \frac{1}{\sqrt{n}} - \Delta$. Using this inequality and properties of the hyperbolic secant we obtain

$$h_2 \leq 4 \exp(-2\frac{q_j}{\mu} - \log\mu) \leq \exp(\frac{2\Delta}{\mu} - \frac{2}{\mu\sqrt{n}} - \log\mu + \log 4)$$

and plugging in $\mu = \frac{c}{\sqrt{n}\log n}$ for some $c < 1$

$$\leq \exp(\frac{2\Delta}{\mu} - \frac{2\log n}{c} - \log c + \frac{1}{2}\log n + \log\log n + \log 4).$$

We can bound this quantity by a constant, say $h_2 \leq \frac{1}{2}$, by requiring

$$A \equiv \frac{2\Delta}{\mu} - \log c + (\frac{1}{2} - \frac{2}{c})\log n + \log\log n \leq -\log 8$$

and for and $c < 1$, using $-\log n + \log\log n < 0$ we have

$$A < \frac{2\Delta}{\mu} - \log c - (\frac{2}{c} - 1)\log n.$$

Since $\Delta$ can be taken arbitrarily small, it is clear that $c$ can be chosen in an $n$-independent manner such that $A \leq -\log 8$. We then find

$$h_1 - h_2 \geq \tanh(\frac{1}{\mu}) - \frac{1}{2} \geq \tanh(\sqrt{n}\log n) - \frac{1}{2} > 0$$

since this inequality is strict, $\Delta$ can be chosen small enough such that $\left|O(\Delta^2)\right| < \Delta(h_1 - h_2)$ and hence

$$h > 0,$$

proving 9.

It follows that under negative gradient flow, a point with $|q_j| < \|\boldsymbol{q}\|_\infty$ cannot flow to a point $\boldsymbol{q}'$ such that $|q_j'| = \|\boldsymbol{q}'\|_\infty$. From the form of the critical points, for every such $j$, $\boldsymbol{q}$ must thus flow to a point such that $q_j' = 0$ (the value of the $j$ coordinate cannot pass through $0$ to a point where $|q_j'| = \|\boldsymbol{q}'\|_\infty$ since from smoothness of the objective this would require passing some $\boldsymbol{q}''$ with $q_j'' = 0$, at which point $\mathrm{grad}\,[f_{\mathrm{Sep}}]\,(\boldsymbol{q}'')_j = 0)$.

As for the maximal magnitude coordinates, if there is more than one coordinate satisfying $|q_{i_1}| = |q_{i_2}| = \|\boldsymbol{q}\|_\infty$, it is clear from symmetry that at any subsequent point $\boldsymbol{q}'$ along the gradient flow line $\left|q_{i_1}'\right| = \left|q_{i_2}'\right|$. These coordinates cannot change sign since from the smoothness of the objective this would require that they pass through a point where they have magnitude smaller than $1/\sqrt{n}$, at which point some other coordinate must have a larger magnitude (in order not to violate the spherical constraint), contradicting the above result for non-maximal elements. It follows that the sign pattern of these elements is preserved during the flow. Thus there is a single critical point to which any $\boldsymbol{q}$ can flow, and this is given by setting all the coordinates with $|q_j| < \|\boldsymbol{q}\|_\infty$ to 0 and multiplying the remaining coordinates by a positive constant to ensure the resulting vector is on $\mathbb{S}^n$. Denoting this critical point by $\boldsymbol{\alpha}$, there is a vector $\boldsymbol{b}$ such that $\boldsymbol{q} = \mathcal{P}_{\mathbb{S}^{n-1}}\,[\boldsymbol{a}(\boldsymbol{\alpha}) + \boldsymbol{b}]$ and $\mathrm{supp}(\boldsymbol{a}(\boldsymbol{\alpha})) \cap \mathrm{supp}(\boldsymbol{b}) = \varnothing$,

$\|\boldsymbol{b}\|_\infty < 1$ with the form of $\boldsymbol{a}(\boldsymbol{\alpha})$ given by 5 . The collection of all such points defines the stable manifold of $\boldsymbol{\alpha}$.

$\square$

**Proof of Lemma 2: (Separable objective gradient projection).** i) We consider the $\text{sign}(w_i) = 1$ case; the $\text{sign}(w_i) = -1$ case follows directly. Recalling that $\boldsymbol{u}^{(i)*}\text{grad}[f_{\text{Sep}}](\boldsymbol{q}(\boldsymbol{w})) = \tanh\left(\frac{w_i}{\mu}\right) - \tanh\left(\frac{q_n}{\mu}\right)\frac{w_i}{q_n}$, we first prove

$$\tanh\left(\frac{w_i}{\mu}\right) - \tanh\left(\frac{q_n}{\mu}\right)\frac{w_i}{q_n} \geq c(q_n - w_i) \tag{10}$$

for some $c > 0$ whose form will be determined later. The inequality clearly holds for $w_i = q_n$. To verify that it holds for smaller values of $w_i$ as well, we now show that

$$\frac{\partial}{\partial w_i}\left[\tanh\left(\frac{w_i}{\mu}\right) - \tanh\left(\frac{q_n}{\mu}\right)\frac{w_i}{q_n} - c(q_n - w_i)\right] < 0$$

which will ensure that it holds for all $w_i$. We define $s^2 = 1 - ||\boldsymbol{w}||^2 + w_i^2$ and denote $q_n = \sqrt{s^2 - w_i^2}$ to extract the $w_i$ dependence, giving

$$\frac{\partial}{\partial w_i}\left[\tanh\left(\frac{w_i}{\mu}\right) - \tanh\left(\frac{q_n}{\mu}\right)\frac{w_i}{q_n} - c(q_n - w_i)\right]$$

$$= \begin{array}{c} \frac{1}{\mu}\text{sech}^2\left(\frac{w_i}{\mu}\right) + \frac{1}{\mu}\text{sech}^2\left(\frac{\sqrt{s^2-w_i^2}}{\mu}\right)\frac{w_i^2}{s^2-w_i^2} \\ -\tanh\left(\frac{\sqrt{s^2-w_i^2}}{\mu}\right)\frac{s^2}{(s^2-w_i^2)^{3/2}} + c(\frac{w_i}{\sqrt{s^2-w_i^2}} + 1) \end{array}$$

$$\leq \begin{array}{c} \frac{4}{\mu}\left(e^{-2\frac{w_i}{\mu}} + e^{-2\frac{\sqrt{s^2-w_i^2}}{\mu}}\right) \\ -\tanh\left(\frac{\sqrt{s^2-w_i^2}}{\mu}\right)\frac{s^2}{(s^2-w_i^2)^{3/2}} + 2c \end{array}$$

Where in the last inequality we used properties of the sech function and $q_n \geq w_i$. We thus want to show

$$\frac{4}{\mu}\left(e^{-2\frac{w_i}{\mu}} + e^{-2\frac{q_n}{\mu}}\right) + 2c \leq \tanh\left(\frac{q_n}{\mu}\right)\frac{q_n^2 + w_i^2}{q_n^3}$$

and using $\log(\frac{1}{\mu})\mu \leq w_i \leq q_n$ and $c = \frac{\frac{1-\mu^2}{1+\mu^2} - 8\mu}{2}$ we have

$$\frac{4}{\mu}\left(e^{-2\frac{w_i}{\mu}} + e^{-2\frac{q_n}{\mu}}\right) + 2c$$

$$\leq \frac{8e^{-2\frac{w_i}{\mu}}}{\mu} + 2c \leq 8\mu + 2c \leq \frac{1-\mu^2}{1+\mu^2}$$

$$= \tanh\left(\log(\frac{1}{\mu})\right) \leq \tanh\left(\frac{q_n}{\mu}\right)\frac{1}{q_n}$$

$$< \tanh\left(\frac{q_n}{\mu}\right)\frac{q_n^2 + w_i^2}{q_n^3}$$

and it follows that 10 holds. For $\mu < \frac{1}{16}$ we are guaranteed that $c > 0$.

From examining the RHS of 10 (and plugging in $q_n = \sqrt{s^2 - w_i^2}$) we see that any lower bound on the gradient of an element $w_j$ applies also to any element $|w_i| \leq |w_j|$. Since for $|w_j| = \|\boldsymbol{w}\|_\infty$ we have $q_n - w_j = w_j \zeta$, for every $\log(\frac{1}{\mu})\mu \leq w_i$ we obtain the bound

$$\boldsymbol{u}^{(i)*}\mathrm{grad}[f_{\mathrm{Sep}}](\boldsymbol{q}(\boldsymbol{w})) \geq c\,\|\boldsymbol{w}\|_\infty\,\zeta$$

$\square$

**Proof of Theorem 1: (Gradient descent convergence rate for separable function).**
We obtain a convergence rate by first bounding the number of iterations of Riemannian gradient descent in $\mathcal{C}_{\zeta_0}\backslash\mathcal{C}_1$, and then considering $\mathcal{C}_1\backslash B_r^\infty$.

From Lemma 16 we obtain $\mathcal{C}_{\zeta_0}\backslash\mathcal{C}_1 \subseteq \mathcal{C}_{\zeta_0}\backslash B_{1/\sqrt{n+3}}^\infty$. Choosing $c_2$ so that $\mu < \frac{1}{2}$, we can apply Lemma 2, and for $\boldsymbol{u}$ defined in 7, we thus have

$$|w_i| > \mu \log(\frac{1}{\mu}) \Rightarrow \boldsymbol{u}^{(i)*}\mathrm{grad}[f_{\mathrm{Sep}}](\boldsymbol{q}(\boldsymbol{w})) > c\|\boldsymbol{w}\|_\infty\zeta_0.$$

Since from Lemma 7 the Riemannian gradient norm is bounded by $\sqrt{n}$, we can choose $c_1, c_2$ such that $\mu \log(\frac{1}{\mu}) < \frac{1}{2\sqrt{n+3}}, \eta < \frac{1}{6\sqrt{n^2+3n}}$. This choice of $\eta$ then satisfies the conditions of Lemma 17 with $r = \mu \log(\frac{1}{\mu}), b = \frac{1}{\sqrt{n+3}}, M = \sqrt{n}$, which gives that after a gradient step

$$\zeta' \geq \zeta\left(1 + \frac{c}{2}\sqrt{\frac{n}{n+3}}\eta\right) \geq \zeta\left(1 + \tilde{c}\eta\right) \tag{11}$$

for some suitably chosen $\tilde{c} > 0$. If we now define by $\boldsymbol{w}^{(t)}$ the $t$-th iterate of Riemannian gradient descent and $\zeta^{(t)} \equiv \frac{q_n^{(t)}}{\|\boldsymbol{w}^{(t)}\|_\infty} - 1, \zeta^{(0)} \equiv \zeta_0$, for iterations such that $\boldsymbol{w}^{(t)} \in \mathcal{C}_\zeta\backslash\mathcal{C}_1$ we find

$$\zeta^{(t)} \geq \zeta^{(t-1)}\left(1 + \tilde{c}\eta\right) \geq \zeta_0\left(1 + \tilde{c}\eta\right)^t$$

and the number of iterations required to exit $\mathcal{C}_{\zeta_0}\backslash\mathcal{C}_1$ is

$$t_1 = \frac{\log(\frac{1}{\zeta_0})}{\log(1 + \tilde{c}\eta)}. \tag{12}$$

To bound the remaining iterations, we use Lemma 2 to obtain that for every $\boldsymbol{w} \in \mathcal{C}_{\zeta_0}\backslash B_r^\infty$,

$$\|\mathrm{grad}[f_{\mathrm{Sep}}](\boldsymbol{q}(\boldsymbol{w}))\|^2 \geq \frac{\left\|\boldsymbol{u}^{(i)*}\mathrm{grad}[f_{\mathrm{Sep}}](\boldsymbol{q}(\boldsymbol{w}))\right\|^2}{\|\boldsymbol{u}^{(i)}\|^2} \geq \zeta_0^2 c^2 r^2$$

where we have used $\|\boldsymbol{u}^{(i)}\|^2 = 1 + \frac{w_i^2}{q_n^2} \leq 2$. We thus have

$$\sum_{i=0}^{T-1}\left\|\mathrm{grad}[f_{\mathrm{Sep}}](\boldsymbol{q}(\boldsymbol{w})^{(i)})\right\|^2$$

$$= \sum_{i=0}^{t_1-1}\left\|\mathrm{grad}[f_{\mathrm{Sep}}](\boldsymbol{q}(\boldsymbol{w})^{(i)})\right\|^2 + \sum_{i=t_1}^{T-1}\left\|\mathrm{grad}[f_{\mathrm{Sep}}](\boldsymbol{q}(\boldsymbol{w})^{(i)})\right\|^2$$

$$> \frac{\zeta_0^2 c^2}{(n+3)}t_1 + (T - t_1)c^2 r^2. \tag{13}$$

Choosing $\eta < \frac{1}{2L}$ where $L$ is the gradient Lipschitz constant of $f_s$, from Lemma 5 we obtain

$$\frac{2\left(f_{\mathrm{Sep}}(\boldsymbol{q}^{(0)}) - f_{\mathrm{Sep}}^*\right)}{\eta} > \sum_{i=0}^{T-1}\left\|\mathrm{grad}[f_{\mathrm{Sep}}](\boldsymbol{q}^{(i)})\right\|^2.$$

According to Lemma B, $L = 1/\mu$ and thus the above holds if we demand $\eta < \frac{\mu}{2}$. Combining 12 and 13 gives

$$T < \frac{2\left(f_{\text{Sep}}(\boldsymbol{q}^{(0)}) - f_{\text{Sep}}^*\right)}{\eta c^2 r^2} + \frac{\left(1 - \frac{\zeta_0^2}{(n+3)r^2}\right)\log(\frac{1}{\zeta_0})}{\log(1 + \tilde{c}\eta)}.$$

To obtain the final rate, we use in $g(\boldsymbol{w}^0) - g^* \leq \sqrt{n}$ and $\tilde{c}\eta < 1 \Rightarrow \frac{1}{\log(1+\tilde{c}\eta)} < \frac{\tilde{C}}{\tilde{c}\eta}$ for some $\tilde{C} > 0$. Thus one can choose $C > 0$ such that

$$T < \frac{C}{\eta}\left(\frac{\sqrt{n}}{r^2} + \log(\frac{1}{\zeta_0})\right). \tag{14}$$

From Lemma 1 the ball $B_r^\infty$ contains a global minimizer of the objective, located at the origin.

The probability of initializing in $\bigcup_{\tilde{A}} \mathcal{C}_{\zeta_0}$ is simply given from Lemma 3 and by summing over the $2n$ possible choices of $\mathcal{C}_{\zeta_0}$, one for each global minimizer (corresponding to a single signed basis vector).

$\square$

**Lemma 5** (Riemannian gradient descent iterate bound)**.** *For a Riemannian gradient descent algorithm on the sphere with step size $t_k < \frac{1}{2L}$, where $L$ is a lipschitz constant for $\nabla f(\boldsymbol{q})$, one has*

$$f(\boldsymbol{q}_1) - f(\boldsymbol{q}^\star) \geq f(\boldsymbol{q}_1) - f(\boldsymbol{q}_T)$$
$$\geq \frac{t_k}{2}\|\text{grad}\,[f]\,(\boldsymbol{q}_k)\|^2.$$

*Proof.* Just as in the euclidean setting, we can obtain a lower bound on progress in function values of iterates of the Riemannian gradient descent algorithm from a lower bound on the Riemannian gradient. Consider $f : S^{n-1} \to \mathbb{R}$, which has $L$-lipschitz gradient. Let $\boldsymbol{q}_k$ denote the current iterate of Riemannian gradient descent, and let $t_k > 0$ denote the step size. Then we can form the Taylor approximation to $f \circ \text{Exp}_{\boldsymbol{q}_k}(\boldsymbol{v})$ at $\boldsymbol{0}_{\boldsymbol{q}_k}$:

$$\hat{f} : B_1(\boldsymbol{0}_{\boldsymbol{q}_k}) \cap T_{\boldsymbol{q}_k} S^{n-1} \to \mathbb{R} : \boldsymbol{v} \mapsto f(\boldsymbol{q}_k) + \langle \boldsymbol{v}, \nabla f(\boldsymbol{q}_k)\rangle.$$

From Taylor's theorem, we have for any $\boldsymbol{v} \in B_1(\boldsymbol{0}_{\boldsymbol{q}_k}) \cap T_{\boldsymbol{q}_k} S^{n-1}$

$$|\hat{f}(\boldsymbol{v}) - f \circ \text{Exp}_{\boldsymbol{q}_k}(\boldsymbol{v})| \leq \frac{1}{2}\|\text{Hess}[f](\boldsymbol{q}_k)\|\|\boldsymbol{v} - \boldsymbol{0}_{\boldsymbol{q}_k}\|^2,$$

where the matrix norm is the operator norm on $\mathbb{R}^{n \times n}$. Using the gradient-lipschitz property of $f$, we readily compute

$$\|\text{Hess}[f](\boldsymbol{q}_k)\| \leq \|\nabla^2 f(\boldsymbol{q}_k)\| + |\langle \nabla f(\boldsymbol{q}_k), \boldsymbol{q}_k\rangle|$$
$$\leq 2L,$$

since $\nabla f(\boldsymbol{0}) = \boldsymbol{0}$ and $\boldsymbol{q}_k \in S^{n-1}$. We thus have

$$f \circ \text{Exp}_{\boldsymbol{q}_k}(\boldsymbol{v}) \leq f(\boldsymbol{q}_k) + \langle \boldsymbol{v}, \nabla f(\boldsymbol{q}_k)\rangle + L\|\boldsymbol{v}\|^2.$$

If we put $\boldsymbol{v} = -t_k\text{grad}[f](\boldsymbol{q}_k)$ and write $\boldsymbol{q}_{k+1} = \text{Exp}_{\boldsymbol{q}_k}(-t_k\text{grad}\,[f]\,(\boldsymbol{q}_k))$, the previous expression becomes

$$f(\boldsymbol{q}_{k+1}) \leq f(\boldsymbol{q}_k) - t_k\|\text{grad}\,[f]\,(\boldsymbol{q}_k)\|^2 + t_k^2 L\|\text{grad}\,[f]\,(\boldsymbol{q}_k)\|^2$$
$$\leq f(\boldsymbol{q}_k) - \frac{t_k}{2}\|\text{grad}\,[f]\,(\boldsymbol{q}_k)\|^2$$

if $t_k < \frac{1}{2L}$. Thus progress in objective value is guaranteed by lower-bounding the Riemannian gradient.

As in the euclidean setting, summing the previous expression over iterations $k$ now yields

$$\sum_{k=1}^{T-1} f(\boldsymbol{q}_k) - f(\boldsymbol{q}_{k+1}) = f(\boldsymbol{q}_1) - f(\boldsymbol{q}_T)$$

$$\geq \frac{t_k}{2} \sum_{k=1}^{T-1} \|\text{grad}\,[f]\,(\boldsymbol{q}_k)\|^2;$$

in addition, it holds $f(\boldsymbol{q}_1) - f(\boldsymbol{q}_T) \leq f(\boldsymbol{q}_1) - f(\boldsymbol{q}^\star)$. Plugging in a constant step size gives the desired result. $\square$

**Lemma 6** (Lipschitz constant of $\nabla f$). *For any $\boldsymbol{x}_1, \boldsymbol{x}_2 \in \mathbb{R}^n$, it holds*

$$\|\nabla f(\boldsymbol{x}_1) - \nabla f(\boldsymbol{x}_2)\| \leq \frac{1}{\mu}\|\boldsymbol{x}_1 - \boldsymbol{x}_2\|.$$

*Proof.* It will be enough to study a single coordinate function of $\nabla f$. Using a derivative given in section D.1, we have for $x \in \mathbb{R}$

$$\frac{d}{dx}\tanh(x/\mu) = \frac{1}{\mu}\text{sech}^2\left(\frac{x}{\mu}\right).$$

A bound on the magnitude of the derivative of this smooth function implies a lipschitz constant for $x \mapsto \tanh(x/\mu)$. To find the bound, we differentiate again and find the critical points of the function. We have, using the chain rule,

$$\frac{d}{dx}\left(\frac{1}{\mu}\text{sech}^2\left(\frac{x}{\mu}\right)\right) = \frac{-4}{\mu}\text{sech}\left(\frac{x}{\mu}\right)\cdot\frac{1}{(e^{x/\mu}+e^{-x/\mu})^2}$$

$$\cdot\left(\frac{1}{\mu}e^{x/\mu} - \frac{1}{\mu}e^{-x/\mu}\right)$$

$$= -\frac{1}{\mu^2}\frac{e^{x/\mu}-e^{-x/\mu}}{(e^{x/\mu}+e^{-x/\mu})^3}.$$

The denominator of this final expression vanishes nowhere. Hence, the only critical point satisfies $x/\mu = -x/\mu$, which implies $x = 0$. Therefore it holds

$$\frac{d}{dx}\tanh(x/\mu) \leq \frac{1}{\mu}\text{sech}^2(0) = \frac{1}{\mu},$$

which shows that $\tanh(x/\mu)$ is $(1/\mu)$-lipschitz.

Now let $\boldsymbol{x}_1$ and $\boldsymbol{x}_2$ be any two points of $\mathbb{R}^n$. Then one has

$$\|\nabla f(\boldsymbol{x}_1) - \nabla f(\boldsymbol{x}_2)\| = \left(\sum_i \left(\tanh(x_{1i}/\mu) - \tanh(x_{2i}/\mu)\right)^2\right)^{1/2}$$

$$= \left(\sum_i |\tanh(x_{1i}/\mu) - \tanh(x_{2i}/\mu)|^2\right)^{1/2}$$

$$\leq \left(\sum_i \frac{1}{\mu}\left|\frac{x_{1i}}{\mu} - \frac{x_{2i}}{\mu}\right|^2\right)^{1/2}$$

$$= \frac{1}{\mu}\|\boldsymbol{x}_1 - \boldsymbol{x}_2\|,$$

completing the proof. $\square$

**Lemma 7** (Separable objective gradient bound). *The separable objective gradient obeys*

$$\|\nabla_{\boldsymbol{w}}g(\boldsymbol{w})\| \leq \sqrt{2n}$$

$$\|\text{grad}[f](\boldsymbol{q})\| \leq \sqrt{n}$$

*Proof.* Recalling that the Euclidean gradient is given by $\nabla f_{\text{Sep}}(\boldsymbol{q})_i = \tanh\left(\frac{q_i}{\mu}\right)$ we use Jensen's inequality, convexity of the $L^2$ norm and the triangle inequality to obtain

$$\|\nabla g_s(\boldsymbol{w})\|^2 \le \|\nabla f_{\text{Sep}}(\boldsymbol{q})\|^2 + \left|\tanh\left(\frac{q_n}{\mu}\right)\right|^2 \frac{\|\boldsymbol{w}\|^2}{q_n^2} \le 2n$$

while

$$\|\text{grad}[f_{\text{Sep}}](\boldsymbol{q})\| = \|(\boldsymbol{I} - qq^*)\nabla f_{\text{Sep}}(\boldsymbol{q})\| \le \|\nabla f_{\text{Sep}}(\boldsymbol{q})\| = \sqrt{n}$$

$\square$

## B  Proofs - Dictionary Learning

**Proof of Lemma 4:(Dictionary learning population gradient).** For simplicity we consider the case $\text{sign}(\boldsymbol{w}_i) = 1$. The converse follows by a similar argument. We have

$$\boldsymbol{u}^{(i)*}\text{grad}[f_{DL}^{pop}](\boldsymbol{q}(\boldsymbol{w})) =$$

$$\mathbb{E}_{\boldsymbol{x}}\left[\tanh\left(\frac{\boldsymbol{q}^*(\boldsymbol{w})\boldsymbol{x}}{\mu}\right)\left(-x_n\frac{w_i}{q_n} + x_i\right)\right] \tag{15}$$

Following the notation of (38), we write $x_j = b_j v_j$ where $b_j \sim \text{Bern}(\theta), v_j \sim \mathcal{N}(0,1)$ and denote the vectors of these variables by $\mathcal{J}, v$ respectively. Defining $Y^{(n)} = \sum_{j \ne n} q(\boldsymbol{w})_j x_j, X^{(n)} = q_n v_n$, $Y$ is Gaussian conditioned on a certain setting of $\mathcal{J}$. Using Lemma 40 in (38) the first term in 15 is

$$-\frac{w_i\theta}{q_n^2}\mathbb{E}_{\boldsymbol{v},\mathcal{J}|b_n=1}\left[\tanh\left(\frac{Y^{(n)} + X^{(n)}}{\mu}\right)X^{(n)}\right]$$

$$= -\frac{w_i}{\mu}\theta\mathbb{E}_{\boldsymbol{v},\mathcal{J}|b_n=1}\left[\text{sech}^2\left(\frac{Y^{(n)} + X^{(n)}}{\mu}\right)\right]$$

and similarly the second term in 15 is, with $X^{(i)} = w_i v_i, Y^{(i)} = \sum_{j \ne i} q(\boldsymbol{w})_j x_j$

$$\frac{\theta}{w_i}\mathbb{E}_{\boldsymbol{v},\mathcal{J}|b_i=1}\left[\tanh\left(\frac{Y^{(i)} + X^{(i)}}{\mu}\right)X^{(i)}\right]$$

$$= \frac{w_i\theta}{\mu}\mathbb{E}_{\boldsymbol{v},\mathcal{J}|b_i=1}\left[\text{sech}^2\left(\frac{\boldsymbol{q}^*(\boldsymbol{w})\boldsymbol{x}}{\mu}\right)\right]$$

if we now define $X = \sum_{j \ne n, i} q^*(\boldsymbol{w})_j x_j$ we have

$$\boldsymbol{u}^{(i)*}\text{grad}[f_{DL}^{pop}](\boldsymbol{q}(\boldsymbol{w})) =$$

$$= \frac{w_i\theta}{\mu}\left(\begin{array}{c} \mathbb{E}_{\boldsymbol{v},\mathcal{J}|b_i=1}\left[\text{sech}^2\left(\frac{\boldsymbol{q}^*(\boldsymbol{w})\boldsymbol{x}}{\mu}\right)\right] \\ -\mathbb{E}_{\boldsymbol{v},\mathcal{J}|b_n=1}\left[\text{sech}^2\left(\frac{\boldsymbol{q}^*(\boldsymbol{w})\boldsymbol{x}}{\mu}\right)\right] \end{array}\right)$$

$$= \frac{w_i\theta}{\mu}\mathbb{E}_{\boldsymbol{v},\mathcal{J}}\left[\begin{array}{c} \text{sech}^2\left(\frac{X + b_n q_n v_n + w_i v_i}{\mu}\right) \\ -\text{sech}^2\left(\frac{X + q_n v_n + w_i b_i v_i}{\mu}\right) \end{array}\right]$$

$$= \frac{w_i\theta(1-\theta)}{\mu}\mathbb{E}_{\boldsymbol{v},\mathcal{J}\setminus\{n,i\}}\left[\begin{array}{c} \text{sech}^2\left(\frac{X + w_i v_i}{\mu}\right) \\ -\text{sech}^2\left(\frac{X + q_n v_n}{\mu}\right) \end{array}\right] \tag{16}$$

## B.1 Bounds for $\mathbb{E}\left[\text{sech}^2(Y)\right]$

We already have a lower bound in Lemma 20 of (38) that we can use for the second term, so we need an upper bound for the first term. Following from p. 865, we define $Y \sim \mathcal{N}(0, \sigma_Y^2)$, $Z = \exp\left(\frac{-2Y}{\mu}\right)$, and defining $\beta = 1 - \frac{1}{\sqrt{T}}$ for some $T > 1$ we have

$$\text{sech}^2(Y/\mu) = \frac{4Z}{(1+Z)^2} \leq \frac{4Z}{(1+\beta Z)^2} = \sum_{k=0}^{\infty} b_k Z^{k+1}$$

Where $b_k = (-\beta)^k(k+1)$. Using B.3 from Lemma 40 in (38) we have

$$\mathbb{E}\left[\sum_{k=0}^{\infty} b_k Z^{k+1} \mathbb{1}_{Y>0}\right] = \sum_{k=0}^{\infty} b_k \mathbb{E}\left[e^{-2(k+1)Y/\mu} \mathbb{1}_{Y>0}\right]$$

$$= \sum_{k=0}^{\infty} b_k \exp\left(\frac{1}{2}\left(\frac{2(k+1)}{\mu}\right)^2 \sigma_Y^2\right) \Phi^c\left(\frac{2(k+1)}{\mu}\sigma_Y\right)$$

Where $\Phi^c(x)$ is the complementary Gaussian CDF (The exchange of summation and expectation is justified since $Y > 0$ implies $Z \in [0,1]$, see proof of Lemma 18 in (38) for details). Using the following bounds $\frac{1}{\sqrt{2\pi}}\left(\frac{1}{x} - \frac{1}{x^3}\right)e^{-x^2/2} \leq \Phi^c(x) \leq \frac{1}{\sqrt{2\pi}}\left(\frac{1}{x} - \frac{1}{x^3} + \frac{3}{x^5}\right)e^{-x^2/2}$ by applying the upper (lower) bound to the even (odd) terms in the sum, and then adding a non-negative quantity, we obtain

$$\leq \frac{1}{\sqrt{2\pi}}\sum_{k=0}^{\infty}(-\beta)^k(k+1)\left(\frac{1}{\frac{2(k+1)}{\mu}\sigma_Y} - \frac{1}{\left(\frac{2(k+1)}{\mu}\sigma_Y\right)^3}\right)$$

$$+ \frac{1}{\sqrt{2\pi}}\sum_{k=0}^{\infty}\beta^k(k+1)\left(\frac{3}{\left(\frac{2(k+1)}{\mu}\sigma_Y\right)^5}\right)$$

and using $\sum_{k=0}^{\infty}(-\beta)^k = \frac{1}{1+\beta}$, $\sum_{k=0}^{\infty}\frac{b_k}{(k+1)^3} \geq 0$, $\sum_{k=0}^{\infty}\frac{|b_k|}{(k+1)^5} \leq 2$ (from Lemma 17 in (38)) and taking $T \to \infty$ so that $\beta \to 1$ we have

$$\sum_{k=0}^{\infty} b_k \mathbb{E}\left[Z^{k+1}\mathbb{1}_{Y>0}\right] \leq \frac{1}{2\sqrt{2\pi}}\frac{1}{\frac{2}{\mu}\sigma_Y} + \frac{1}{\sqrt{2\pi}}\frac{6}{\left(\frac{2}{\mu}\sigma_Y\right)^5}$$

giving the upper bound

$$\mathbb{E}\left[\text{sech}^2(Y/\mu)\right] = \mathbb{E}\left[1 - \tanh^2(Y/\mu)\right] \leq 8\sum_{k=0}^{\infty} b_k \mathbb{E}\left[Z^{k+1}\mathbb{1}_{Y>0}\right]$$

$$\leq \sqrt{\frac{2}{\pi}}\frac{\mu}{\sigma_Y} + \frac{3\mu^5}{2\sqrt{2\pi}\sigma_Y^5}$$

while the lower bound (Lemma 20 in (38)) is

$$\sqrt{\frac{2}{\pi}}\frac{\mu}{\sigma_Y} - \frac{2\mu^3}{\sqrt{2\pi}\sigma_Y^3} - \frac{3\mu^5}{2\sqrt{2\pi}\sigma_Y^5} \leq \mathbb{E}\left[\text{sech}^2(Y)\right]$$

## B.2 Gradient bounds

After conditioning on $\mathcal{J}\backslash\{n,i\}$ the variables $X + q_n v_n, X + q_i v_i$ are Gaussian. We can thus plug the bounds into 16 to obtain

$$\boldsymbol{u}^{(i)*}\mathrm{grad}[f_{DL}^{pop}](\boldsymbol{q}(\boldsymbol{w})) \geq \sqrt{\frac{2}{\pi}}w_i\theta(1-\theta)$$

$$*\mathbb{E}_{\mathcal{J}\backslash\{n,i\}}\left[ \begin{array}{c} \frac{1}{\sqrt{\sigma_X^2+w_i^2}} - \frac{\mu^2}{\left(\sigma_X^2+w_i^2\right)^{3/2}} - \frac{3\mu^4}{4\left(\sigma_X^2+w_i^2\right)^{5/2}} \\ -\frac{1}{\sqrt{\sigma_X^2+q_n^2}} - \frac{3\mu^4}{4\left(\sigma_X^2+q_n^2\right)^{5/2}} \end{array} \right]$$

$$\geq \sqrt{\frac{2}{\pi}}w_i\theta(1-\theta)\left( \begin{array}{c} \mathbb{E}_{\mathcal{J}\backslash\{n,i\}}\left[ \frac{\sqrt{\sigma_X^2+q_n^2}-\sqrt{\sigma_X^2+w_i^2}}{\sqrt{\sigma_X^2+q_n^2}\sqrt{\sigma_X^2+w_i^2}} \right] \\ -\frac{\mu^2}{w_i^3} - \frac{3\mu^4}{2w_i^5} \end{array} \right)$$

the term in the expectation is positive since $q_n > ||w||_\infty (1 + \zeta) > w_i$ giving

$$\geq \sqrt{\frac{2}{\pi}}w_i\theta(1-\theta)\left( \begin{array}{c} \mathbb{E}_{\mathcal{J}\backslash\{n,i\}}\left[ \begin{array}{c} \sqrt{\sigma_X^2+q_n^2} \\ -\sqrt{\sigma_X^2+w_i^2} \end{array} \right] \\ -\frac{\mu^2}{w_i^3} - \frac{3\mu^4}{2w_i^5} \end{array} \right)$$

. To extract the $\zeta$ dependence we plug in $q_n > w_i(1 + \zeta)$ and develop to first order in $\zeta$ (since the resulting function of $\zeta$ is convex) giving

$$\geq \sqrt{\frac{2}{\pi}}w_i\theta(1-\theta)\left( \begin{array}{c} \mathbb{E}_{\mathcal{J}\backslash\{n,i\}}\left[ \frac{w_i^2\zeta}{\sqrt{\sigma_X^2+w_i^2}} \right] \\ -\frac{\mu^2}{w_i^3} - \frac{3\mu^4}{2w_i^5} \end{array} \right)$$

$$\geq \sqrt{\frac{2}{\pi}}\theta(1-\theta)\left( w_i^3\zeta - \frac{\mu^2}{w_i^2} - \frac{3\mu^4}{2w_i^4} \right)$$

Given some $\zeta$ and $r$ such that $w_i > r$, if we now choose $\mu$ such that $\mu < \sqrt{\frac{\sqrt{1+\frac{3}{4}r^3\zeta}-1}{3}}r$ we have the desired result. This can be achieved by requiring $\mu < c_1 r^{5/2}\sqrt{\zeta}$ for a suitably chosen $c_1 > 0$.

$\square$

**Lemma 8** (Point-wise concentration of projected gradient). *For $\boldsymbol{u}^{(i)}$ defined in 7, the gradient of the objective 1 obeys*

$$\mathbb{P}\left[ \left| \boldsymbol{u}^{(i)*}\mathrm{grad}[f_{DL}](\boldsymbol{q}) - \mathbb{E}\left[ \boldsymbol{u}^{(i)*}\mathrm{grad}[f_{DL}](\boldsymbol{q}) \right] \right| \geq t \right]$$

$$\leq 2\exp\left( -\frac{pt^2}{4 + 2\sqrt{2}t} \right)$$

**Proof of Lemma 8: (Point-wise concentration of projected gradient).** If we denote by $\boldsymbol{x}^i$ a column of the data matrix with entries $x_j^i \sim BG(\theta)$, we have

$$\boldsymbol{u}^{(i)*}\mathrm{grad}[f_{DL}](\boldsymbol{q}(\boldsymbol{w}))$$

$$= \frac{1}{p}\sum_{k=1}^{p}\tanh\left( \frac{\boldsymbol{q}^*(\boldsymbol{w})\boldsymbol{x}^k}{\mu} \right)\left( x_i^k - x_n^k\frac{w_i}{q_n} \right) \equiv \frac{1}{p}\sum_{k=1}^{p}Z_k$$

. Since $\tanh(x)$ is bounded by 1,

$$|Z_k| \leq \left|\left(x_i^k - x_n^k \frac{w_i}{q_n}\right)\right| \equiv |u^T x^k|$$

. Invoking Lemma 21 from (38) and $\|u\|^2 = 1 + \frac{w_i^2}{q_n^2} \leq 2$ we obtain

$$\mathbb{E}\left[|Z_k|^m\right] \leq \mathbb{E}_{Z \sim \mathcal{N}(0,2)}\left[|Z|^m\right] \leq \sqrt{2}^m (m-1)!!$$

$$\leq 2\sqrt{2}^{m-2} \frac{m!}{2}$$

and using Lemma 36 in (38) with $R = \sqrt{2}, \sigma = \sqrt{2}$ we have

$$\mathbb{P}\left[|\nabla g_{DL}(\boldsymbol{w})_i - \mathbb{E}\left[\nabla g_{DL}(\boldsymbol{w})_i\right]| \geq t\right]$$

$$\leq 2\exp\left(-\frac{pt^2}{4 + 2\sqrt{2}t}\right)$$

$\square$

**Lemma 9** (Projection Lipschitz Constant). *The Lipschitz constant for* $\boldsymbol{u}^{(i)*}\mathrm{grad}[f_{DL}](\boldsymbol{q}(\boldsymbol{w}))$ *is*

$$L = 2\sqrt{n}\,\|\boldsymbol{X}\|_\infty \left(\frac{\|\boldsymbol{X}\|_\infty}{\mu} + 1\right)$$

**Proof of Lemma 9: (Projection Lipschitz Constant).** We have

$$|\boldsymbol{u}^{(j)*}\mathrm{grad}[f_{DL}](\boldsymbol{q}(\boldsymbol{w})) - \boldsymbol{u}^{(j)*}\mathrm{grad}[f_{DL}](\boldsymbol{q}(\boldsymbol{w}'))|$$

$$= \left|\frac{1}{p}\sum_{i=1}^{p}\left[\begin{array}{c}\tanh(\frac{\boldsymbol{q}^*(\boldsymbol{w})\boldsymbol{x}^i}{\mu})\left(x_j^i - \frac{x_n^i}{q_n(\boldsymbol{w})}w_j\right) \\ -\tanh(\frac{\boldsymbol{q}^*(\boldsymbol{w}')\boldsymbol{x}^i}{\mu})\left(x_j^i - \frac{x_n^i}{q_n(\boldsymbol{w}')}w_j'\right)\end{array}\right]\right|$$

$$\equiv \left|\frac{1}{p}\sum_{i=1}^{p}\left[\tanh(\frac{\boldsymbol{q}^*(\boldsymbol{w})\boldsymbol{x}^i}{\mu})s(\boldsymbol{w}) - \tanh(\frac{\boldsymbol{q}^*(\boldsymbol{w}')\boldsymbol{x}^i}{\mu})s(\boldsymbol{w}')\right]\right|$$

where we have defined $s(\boldsymbol{w}) = x_j^i - \frac{x_n}{q_n(\boldsymbol{w})}w_j$. Using $\boldsymbol{q}(\boldsymbol{w}), \boldsymbol{q}(\boldsymbol{w}') \in C \Rightarrow q_n(\boldsymbol{w}), q_n(\boldsymbol{w}') \geq \frac{1}{2\sqrt{n}}$ we have

$$|s(\boldsymbol{w}) - s(\boldsymbol{w}')| = \left|x_n^i\right|\left|\frac{w_j}{q_n(\boldsymbol{w})} - \frac{w_j'}{q_n(\boldsymbol{w}')}\right|$$

$$\leq |x_n|\,2\sqrt{n}\,\|\boldsymbol{w} - \boldsymbol{w}'\|$$

Lemma 25 in (38) gives

$$\left|\tanh(\frac{\boldsymbol{q}^*(\boldsymbol{w})\boldsymbol{x}}{\mu}) - \tanh(\frac{\boldsymbol{q}^*(\boldsymbol{w}')\boldsymbol{x}}{\mu})\right| \leq \frac{2\sqrt{n}}{\mu}\,\|x\|\,\|\boldsymbol{w} - \boldsymbol{w}'\|$$

We also use the fact that tanh is bounded by 1 and $s(\boldsymbol{w})$ is bounded by $\|\boldsymbol{X}\|_\infty$. We can then use Lemma 23 in (38) to obtain

$$|\boldsymbol{u}^{(j)*}\mathrm{grad}[f_{DL}](\boldsymbol{q}(\boldsymbol{w})) - \boldsymbol{u}^{(j)*}\mathrm{grad}[f_{DL}](\boldsymbol{q}(\boldsymbol{w}'))|$$

$$\leq \frac{2\sqrt{n}}{p}\sum_{i=1}^{p}(\frac{1}{\mu}\,\left\|x^i\right\|_\infty^2 + \left\|x^i\right\|_\infty)\,\|\boldsymbol{w} - \boldsymbol{w}'\|$$

$$\leq 2\sqrt{n}\,\|\boldsymbol{X}\|_{\infty}\left(\frac{\|\boldsymbol{X}\|_{\infty}}{\mu}+1\right)\|\boldsymbol{w}-\boldsymbol{w}'\|$$

we thus have $L = 2\sqrt{n}\,\|\boldsymbol{X}\|_{\infty}\left(\frac{\|\boldsymbol{X}\|_{\infty}}{\mu}+1\right)$. $\qquad\qquad\qquad\qquad\qquad\qquad\square$

**Lemma 10** (Uniformized gradient fluctuations). *For all $\boldsymbol{w}\in\mathcal{C}_{\zeta}, i\in[n]$, with probability $\mathbb{P} > \mathbb{P}_y$*

*we have*

$$\left|\begin{array}{c}\boldsymbol{u}^{(i)*}\mathrm{grad}[f_{DL}](\boldsymbol{q}(\boldsymbol{w}))\\-\mathbb{E}\left[\boldsymbol{u}^{(i)*}\mathrm{grad}[f_{DL}](\boldsymbol{q}(\boldsymbol{w}))\right]\end{array}\right|\leq y(\theta,\zeta)$$

*where*

$$\mathbb{P}_y \equiv 2\exp\left(\begin{array}{c}-\frac{1}{4}\frac{py(\theta,\zeta)^2}{4+\sqrt{2}y(\theta,\zeta)}+\log(n)\\+n\log\left(\frac{48\sqrt{n}\left(\frac{4\log(np)}{\mu}+\sqrt{\log(np)}\right)}{y(\theta,\zeta)}\right)\end{array}\right)$$

Proof: B

**Proof of Lemma 10:(Uniformized gradient fluctuations).** For $\boldsymbol{X}\in\mathbb{R}^{n\times p}$ with i.i.d. $BG(\theta)$ entries, we define the event $\mathcal{E}_{\infty}\equiv\{1\leq\|\boldsymbol{X}\|_{\infty}\leq 4\sqrt{\log(np)}\}$. We have

$$\mathbb{P}[\mathcal{E}_{\infty}^c]\leq\theta(np)^{-7}+e^{-0.3\theta np}$$

For any $\varepsilon\in(0,1)$ we can construct an $\varepsilon$-net $N$ for $\mathcal{C}_{\zeta}\backslash B^2_{1/20\sqrt{5(n-1)}}(0)$ with at most $(3/\varepsilon)^n$ points. Using Lemma 9, on $\mathcal{E}_{\infty}$, $\mathrm{grad}[f_{DL}](\boldsymbol{q})_i$ is $L$-Lipschitz with

$$L = 8\sqrt{n}\left(\frac{4\log(np)}{\mu}+\sqrt{\log(np)}\right)$$

. If we choose $\varepsilon = \frac{y(\theta,\zeta)}{2L}$ we have

$$|N|\leq(\frac{6L}{y(\theta,\zeta)})^n$$

. We then denote by $\mathcal{E}_g$ the event

$$\max_{\boldsymbol{w}\in N, i\in[n]}\left|\begin{array}{c}\boldsymbol{u}^{(i)*}\mathrm{grad}[f_{DL}](\boldsymbol{q}(\boldsymbol{w}))\\-\mathbb{E}\left[\boldsymbol{u}^{(i)*}\mathrm{grad}[f_{DL}](\boldsymbol{q}(\boldsymbol{w}))\right]\end{array}\right|\leq\frac{y(\theta,\zeta)}{2}$$

and obtain that on $\mathcal{E}_g\cap\mathcal{E}_{\infty}$

$$\sup_{\boldsymbol{w}\in\mathcal{C}_{\zeta}, i\in[n]}|\nabla g_{DL}(\boldsymbol{w})_i-\mathbb{E}\left[\nabla g_{DL}(\boldsymbol{w})_i\right]|\leq y(\theta,\zeta)$$

. Setting $t=\frac{b(\theta)}{2}$ in the result of Lemma 8 gives that for all $\boldsymbol{w}\in\mathcal{C}_{\zeta}, i\in[n]$,

$$\mathbb{P}\left[\left|\begin{array}{c}\boldsymbol{u}^{(i)*}\mathrm{grad}[f_{DL}](\boldsymbol{q}(\boldsymbol{w}))\\-\mathbb{E}\left[\boldsymbol{u}^{(i)*}\mathrm{grad}[f_{DL}](\boldsymbol{q}(\boldsymbol{w}))\right]\end{array}\right|\geq\frac{y(\theta,\zeta)}{2}\right]$$

$$\leq 2\exp\left(-\frac{1}{4}\frac{py(\theta,\zeta)^2}{4+2\sqrt{2}y(\theta,\zeta)}\right)$$

and thus

$$\mathbb{P}\left[\mathcal{E}_g^c\right]\leq 2\exp\left(\begin{array}{c}-\frac{1}{4}\frac{py(\theta,\zeta)^2}{4+\sqrt{2}y(\theta,\zeta)^2}\\+n\log\left(\frac{6L}{b(\theta)}\right)+\log(n)\end{array}\right)$$

$$\square$$

**Lemma 11** (Gradient descent convergence rate for dictionary learning - population). *For any $1 > \zeta_0 > 0$ and $s > \frac{\mu}{4\sqrt{2}}$, Riemannian gradient descent with step size $\eta < \frac{c_2 s}{n}$ on the dictionary learning population objective 8 with $\mu < \frac{c_4\sqrt{\zeta_0}}{n^{5/4}}, \theta \in (0, \frac{1}{2})$, enters a ball of radius $c_3 s$ from a target solution in*

$$T < \frac{C_1}{\eta\theta}\left(\frac{1}{s} + n\log\frac{1}{\zeta_0}\right)$$

*iterations with probability*

$$\mathbb{P} \geq 1 - 2\log(n)\zeta_0$$

*where the $c_i, C_i$ are positive constants.*

**Proof of Lemma 11: (Gradient descent convergence rate for dictionary learning - population).**
The rate will be obtained by splitting $\mathcal{C}_{\zeta_0}$ into three regions. We consider convergence to $B_s^2(0)$ since this set contains a global minimizer. Note that the balls in the proof are defined with respect to $\boldsymbol{w}$.

**B.3** $\quad \mathcal{C}_{\zeta_0} \backslash B_{1/20\sqrt{5}}^2(0)$

The analysis in this region is completely analogous to that in the first part of the proof of Lemma 1. For every point in this set we have

$$\|\boldsymbol{w}\|_\infty > \frac{1}{20\sqrt{5(n-1)}}$$

. From Lemma 16 we know that $\sqrt{\frac{n-1}{(2+\zeta^{(t)})\zeta^{(t)}+n}} < \frac{1}{20\sqrt{5}} \Rightarrow \boldsymbol{w}^{(t)} \in B_{1/20\sqrt{5}}^2(0)$ hence in this set $\zeta < 8$. If we choose $r = \frac{1}{40\sqrt{5(n-1)}}$, since for every point in this region $r^3\zeta < 1$, we have $\frac{r^{5/2}\sqrt{\zeta}}{2\sqrt{3}} < \sqrt{\frac{\sqrt{1+\frac{3}{4}r^3\zeta}-1}{3}}r = z(r,\zeta)$ and we thus demand $\mu < \frac{\sqrt{\zeta_0}}{\left(40\sqrt{5(n-1)}\right)^{5/2}2\sqrt{3}} \leq \frac{r^{5/2}\sqrt{\zeta}}{2\sqrt{3}}$ and obtain from Lemma 4 that for $|w_i| > r$

$$\boldsymbol{u}^{(i)*}\text{grad}[f_{DL}^{pop}](\boldsymbol{q}(\boldsymbol{w})) \geq \frac{c_{DL}}{(8000(n-1))^{3/2}}$$

. We now require $\eta < \frac{1}{360\sqrt{5\theta n(n-1)}} = \frac{b-r}{3M}$ we can apply Lemma 17 with $b = \frac{1}{20\sqrt{5(n-1)}}, r = \frac{1}{40\sqrt{5(n-1)}}, M = \sqrt{\theta n}$ (since the maximal norm of the Riemannian gradient is $\sqrt{\theta n}$ from Lemma 12), obtaining that at every iteration in this region

$$\zeta' \geq \zeta\left(1 + \frac{\sqrt{n}c_{DL}}{2(8000(n-1))^{3/2}}\eta\right)$$

and the maximal number of iterations required to obtain $\zeta > 8$ and exit this region is given by

$$t_1 = \frac{\log(8/\zeta_0)}{\log\left(1 + \frac{\sqrt{n}c_{DL}}{2(8000(n-1))^{3/2}}\eta\right)} \tag{17}$$

**B.4** $\quad B_{1/20\sqrt{5}}^2(0) \backslash B_s^2(0)$

According to Proposition 7 in (38), which we can apply since $s \geq \frac{\mu}{4\sqrt{2}}, \mu < \frac{9}{50}$, in this region we have

$$\frac{\boldsymbol{w}^*\nabla_{\boldsymbol{w}}g_{DL}^{pop}(\boldsymbol{w})}{\|\boldsymbol{w}\|} \geq c\theta$$

A simple calculation shows that $\nabla_{\boldsymbol{w}}g_{DL}^{pop}(\boldsymbol{w}) = \left(\frac{\partial\boldsymbol{\varphi}}{\partial\boldsymbol{w}}\right)^*\text{grad}[f_{DL}^{pop}](\boldsymbol{q}(\boldsymbol{w}))$ where $\boldsymbol{\varphi}$ is the map defined in 3, and thus

$$\frac{\boldsymbol{w}^* \left(\frac{\partial \varphi}{\partial \boldsymbol{w}}\right)^* \operatorname{grad}[f_{DL}^{pop}](\boldsymbol{q}(\boldsymbol{w}))}{\|\boldsymbol{w}\|} = \frac{\left(\begin{array}{c} \boldsymbol{w}^* \\ -\frac{\|\boldsymbol{w}\|^2}{q_n} \end{array}\right) \operatorname{grad}[f_{DL}^{pop}](\boldsymbol{q}(\boldsymbol{w}))}{\|\boldsymbol{w}\|}$$

$$> \theta c \qquad (18)$$

. Defining $h(\boldsymbol{q}) = \frac{\|\boldsymbol{w}\|^2}{2}$, and denoting by $\boldsymbol{q}'$ an update of Riemannian gradient descent with step size $\eta$, we have (using a Lagrange remainder term)

$$h(\boldsymbol{q}') = h(\boldsymbol{q}) + \frac{\partial h(\boldsymbol{q}')}{\partial \eta}\eta + \underbrace{\int_0^\eta dt \frac{\partial^2 h(\boldsymbol{q}')}{\partial \eta^2}\bigg|_{\eta=t}(\eta - t)}_{\equiv R}$$

$$= \frac{\|\boldsymbol{w}\|^2}{2} - \left\langle \operatorname{grad}[f_{DL}^{pop}](\boldsymbol{q}), \frac{\partial h(\boldsymbol{q})}{\partial \boldsymbol{q}} \right\rangle + R$$

where in the last line we used $\boldsymbol{q}' = \cos(g\eta)\boldsymbol{q} - \sin(g\eta)\frac{\operatorname{grad}[f_{DL}^{pop}](\boldsymbol{q})}{g}$ where $g \equiv \|\operatorname{grad}[f_{DL}^{pop}](\boldsymbol{q})\|$. Since $\left\langle \operatorname{grad}[f_{DL}^{pop}](\boldsymbol{q}), \frac{\partial h(\boldsymbol{q})}{\partial \boldsymbol{q}} \right\rangle = \left\langle \operatorname{grad}[f_{DL}^{pop}](\boldsymbol{q}), (\boldsymbol{I} - \boldsymbol{q}\boldsymbol{q}^*) \frac{\partial h(\boldsymbol{q})}{\partial \boldsymbol{q}} \right\rangle$ and

$$(\boldsymbol{I} - \boldsymbol{q}\boldsymbol{q}^*)\frac{\partial h(\boldsymbol{q})}{\partial \boldsymbol{q}} = (\boldsymbol{I} - \boldsymbol{q}\boldsymbol{q}^*)\left(\begin{array}{c}\boldsymbol{w} \\ -q_n\end{array}\right)$$

$$= \left(\begin{array}{c}\boldsymbol{w} \\ -q_n\end{array}\right) - (\|\boldsymbol{w}\|^2 - q_n^2)\boldsymbol{q} = 2(1 - \|\boldsymbol{w}\|^2)\left(\begin{array}{c}\boldsymbol{w} \\ -\frac{\|\boldsymbol{w}\|^2}{q_n}\end{array}\right)$$

we obtain (using 18)

$$\frac{\|\boldsymbol{w}'\|^2}{2} = \frac{\|\boldsymbol{w}\|^2}{2} + 2(1 - \|\boldsymbol{w}\|^2)\eta\left\langle \operatorname{grad}[f_{DL}^{pop}](\boldsymbol{q}), \left(\begin{array}{c}\boldsymbol{w} \\ -\frac{\|\boldsymbol{w}\|^2}{q_n}\end{array}\right)\right\rangle + R$$

$$< \frac{\|\boldsymbol{w}\|^2}{2} - 2(1 - \|\boldsymbol{w}\|^2)\|\boldsymbol{w}\|\theta c\eta + R$$

It remains to bound $R$. Denoting $r = \left(\begin{array}{c}\boldsymbol{w} \\ -q_n\end{array}\right)^* \operatorname{grad}[f](\boldsymbol{q})$ we have

$$\frac{\partial^2 h(\boldsymbol{q}')}{\partial \eta^2}\bigg|_{\eta=t} = \left(\frac{\partial \boldsymbol{q}'}{\partial \eta}\right)^* \frac{\partial^2 h(\boldsymbol{q})}{\partial \boldsymbol{q}\partial \boldsymbol{q}}\frac{\partial \boldsymbol{q}'}{\partial \eta}\bigg|_{\eta=t} + \frac{\partial h(\boldsymbol{q})}{\partial \boldsymbol{q}}^* \frac{\partial^2 \boldsymbol{q}'}{\partial \eta^2}\bigg|_{\eta=t}$$

$$= \begin{array}{c} \cos^2(gt)\left(\overline{\operatorname{grad}}[f_{DL}^{pop}](\boldsymbol{q})^2 - \operatorname{grad}[f_{DL}^{pop}](\boldsymbol{q})_n^2\right) \\ +g^2\left(\sin^2(gt) - \cos(gt)\right)\left(\|\boldsymbol{w}\|^2 - q_n^2\right) \\ +g\sin(gt)r(1 + 2\cos(gt)) \end{array}$$

hence for some $C > 0$, if $\|\operatorname{grad}[f_{DL}^{pop}](\boldsymbol{q})\| < M$ we have

$$R < CM^2\eta^2$$

and thus choosing $\eta < \frac{(1-\|\boldsymbol{w}\|^2)\|\boldsymbol{w}\|\theta c}{CM^2}$ we find

$$\|\boldsymbol{w}'\|^2 < \|\boldsymbol{w}\|^2 - 2(1 - \|\boldsymbol{w}\|^2)\|\boldsymbol{w}\|c\theta\eta$$

and in our region of interest $\|\boldsymbol{w}'\|^2 < \|\boldsymbol{w}\|^2 - \tilde{c}s\theta\eta$ for some $\tilde{c} > 0$ and thus summing over iterations, we obtain for some $\tilde{C}_2 > 0$

$$t_2 = \frac{\tilde{C}_2}{s\theta\eta}. \tag{19}$$

From Lemma 12, $M = \sqrt{\theta n}$ and thus with a suitably chosen $c_2 > 0$, $\eta < \frac{c_2 s}{n}$ satisfies the above requirement on $\eta$ as well as the previous requirements, since $\theta < 1$.

## B.5   FINAL RATE AND DISTANCE TO MINIMIZER

Combining these results gives, we find that when initializing in $\mathcal{C}_{\zeta_0}$, the maximal number of iterations required for Riemannian gradient descent to enter $B_s^2(0)$ is

$$T \le t_1 + t_2 < \frac{C_1}{\eta\theta}\left(n\log\frac{1}{\zeta_0} + \frac{1}{s}\right)$$

for some suitably chosen $C_1$, where $t_1, t_2$ are given in 17,19. The probability of such an initialization is given by the probability of initializing in one of the $2n$ possible choices of $\mathcal{C}_\zeta$, which is bounded in Lemma 3.

Once $\boldsymbol{w} \in B_s^2(0)$, the distance in $\mathbb{R}^{n-1}$ between $\boldsymbol{w}$ and a solution to the problem (which is a signed basis vector, given by the point $\boldsymbol{w} = \boldsymbol{0}$ or an analog on a different symmetric section of the sphere) is no larger than $s$, which in turn implies that the Riemannian distance between $\varphi(\boldsymbol{w})$ and a solution is no larger than $c_3 s$ for some $c_3 > 0$. We note that the conditions on $\mu$ can be satisfied by requiring $\mu < \frac{c_4\sqrt{\zeta_0}}{n^{5/4}}$.

$\square$

**Lemma 12** (Dictionary learning gradient upper bound). *The dictionary learning population gradient obeys*

$$\|\nabla_{\boldsymbol{w}}g_{DL}^{pop}(\boldsymbol{w})\| \le \sqrt{2\theta n}$$

$$\|\mathrm{grad}[f_{DL}^{pop}](\boldsymbol{q})\| \le \sqrt{\theta n}$$

*while in the finite sample case*

$$\|\nabla_{\boldsymbol{w}}g_{DL}(\boldsymbol{w})\|^2 \le \sqrt{2n}\,\|\boldsymbol{X}\|_\infty$$

$$\|\mathrm{grad}[f_{DL}](\boldsymbol{q})\| \le \sqrt{n}\,\|\boldsymbol{X}\|_\infty$$

*where $\boldsymbol{X}$ is the data matrix with i.i.d. $BG(\theta)$ entries.*

*Proof.* Denoting $\boldsymbol{x} \equiv (\overline{\boldsymbol{x}}, x_n)$ we have

$$\|\nabla_{\boldsymbol{w}}g_{DL}^{pop}(\boldsymbol{w})\|^2 = \left\|\mathbb{E}\left[\tanh\left(\frac{\boldsymbol{q}^*\boldsymbol{x}}{\mu}\right)\left(\overline{\boldsymbol{x}} - x_n\frac{\boldsymbol{w}}{q_n}\right)\right]\right\|^2$$

and using Jensen's inequality, convexity of the $L^2$ norm and the triangle inequality to obtain

$$\le \mathbb{E}\left[\left\|\tanh\left(\frac{\boldsymbol{q}^*\boldsymbol{x}}{\mu}\right)\overline{\boldsymbol{x}}\right\|^2 + \left\|\tanh\left(\frac{\boldsymbol{q}^*\boldsymbol{x}}{\mu}\right)\left(x_n\frac{\boldsymbol{w}}{q_n}\right)\right\|^2\right]$$

$$\le \mathbb{E}\left[\|\overline{\boldsymbol{x}}\|^2 + \left\|x_n\frac{\boldsymbol{w}}{q_n}\right\|^2\right] \le 2\theta n$$

while

$$\|\mathrm{grad}[f_{DL}^{pop}](\boldsymbol{q})\| \le \|\nabla f_{DL}^{pop}(\boldsymbol{q})\|$$

$$= \left\|\mathbb{E}\left[\tanh\left(\frac{\boldsymbol{q}^*\boldsymbol{x}}{\mu}\right)\boldsymbol{x}\right]\right\| \le \sqrt{\theta n}$$

Similarly, in the finite sample size case one obtains

$$\|\nabla_{\boldsymbol{w}} g_{DL}(\boldsymbol{w})\|^2 \leq \frac{1}{p} \sum_{i=1}^{p} \left\|\bar{\boldsymbol{x}}^i\right\|^2 + \left\|x_n^i \frac{\boldsymbol{w}}{q_n}\right\|^2 \leq 2n \left\|\boldsymbol{X}\right\|_\infty^2$$

$$\|\mathrm{grad}[f_{DL}](\boldsymbol{q})\| \leq \frac{1}{p} \sum_{i=1}^{p} \left\|\tanh\left(\frac{\boldsymbol{q}^* \boldsymbol{x}^i}{\mu}\right) \boldsymbol{x}^i\right\|$$

$$\leq \sqrt{n} \left\|\boldsymbol{X}\right\|_\infty$$

□

**Proof of Theorem 2: (Gradient descent convergence rate for dictionary learning).**
The proof will follow exactly that of Lemma 11, with the finite sample size fluctuations decreasing the guaranteed change in $\zeta$ or $\|\boldsymbol{w}\|$ at every iteration (for the initial and final stages respectively) which will adversely affect the bounds.

### B.6 $\mathcal{C}_{\zeta_0} \backslash B^2_{1/20\sqrt{5}}(0)$

To control the fluctuations in the gradient projection, we choose

$$y(\theta, \zeta_0) = \frac{\zeta_0 c_{DL}}{2(8000(n-1))^{3/2}}$$

which can be satisfied by choosing $y(\theta, \zeta_0) = \frac{c_7 \theta (1-\theta) \zeta_0}{n^{3/2}}$ for an appropriate $c_7 > 0$. According to Lemma 10, with probability greater than $\mathbb{P}_y$ we then have

$$\left| \begin{array}{c} \boldsymbol{u}^{(i)*} \mathrm{grad}[f_{DL}](\boldsymbol{q}(\boldsymbol{w})) \\ -\mathbb{E}\left[\boldsymbol{u}^{(i)*} \mathrm{grad}[f_{DL}](\boldsymbol{q}(\boldsymbol{w}))\right] \end{array} \right| \leq y(\theta, \zeta)$$

With the same condition on $\mu$ as in Lemma 11, combined with the uniformized bound on finite sample fluctuations, we have that at every point in this set

$$\boldsymbol{u}^{(i)*} \mathrm{grad}[f_{DL}^{pop}](\boldsymbol{q}(\boldsymbol{w})) \geq \frac{c_{DL}}{2(8000(n-1))^{3/2}}$$

. According to Lemma 12 the Riemannian gradient norm is bounded by $M = \sqrt{n} \left\|\boldsymbol{X}\right\|_\infty$. Choosing $r, b$ as in Lemma 11, we require $\eta < \frac{1}{360\|\boldsymbol{X}\|_\infty \sqrt{5n(n-1)}} = \frac{b-r}{3M}$ and obtain from Lemma 17

$$\zeta' \geq \zeta \left(1 + \frac{\sqrt{n} c_{DL}}{4(8000(n-1))^{3/2}} \eta\right)$$

$$t_1 = \frac{\log(8/\zeta_0)}{\log\left(1 + \frac{\sqrt{n} c_{DL}}{4(8000(n-1))^{3/2}} \eta\right)} \tag{20}$$

### B.7 $B^2_{1/20\sqrt{5}}(0) \backslash B^2_s(0)$

From Theorem 2 in (38) there are numerical constants $c_b, c_\star$ such that in this region

$$\frac{\boldsymbol{w}^* \nabla_{\boldsymbol{w}} g_{DL}(\boldsymbol{w})}{\|\boldsymbol{w}\|} = \frac{\boldsymbol{w}^* \left(\frac{\partial \boldsymbol{\varphi}}{\partial \boldsymbol{w}}\right)^* \mathrm{grad}[f](\boldsymbol{q}(\boldsymbol{w}))}{\|\boldsymbol{w}\|} \geq c_\star \theta$$

with probability $\mathbb{P} > 1 - c_b p^{-6}$. Following the same analysis as in Lemma 11, since from Lemma 12 the norm of the gradient gradient is bounded by $\sqrt{n}\|\boldsymbol{X}\|_\infty$ we require $\eta < \frac{(1-\|\boldsymbol{w}\|^2)\|\boldsymbol{w}\|\theta c_\star}{Cn\|\boldsymbol{X}\|_\infty^2}$ which is satisfied by requiring $\eta < \frac{\tilde{c}\theta s}{n\|\boldsymbol{X}\|_\infty^2}$ for some chosen $\tilde{c} > 0$. We then obtain

$$t_3 = \frac{C_2}{s\theta\eta} \tag{21}$$

for a suitably chosen $C_2 > 0$.

### B.8 FINAL RATE AND DISTANCE TO MINIMIZER

The final bound on the rate is obtained by summing over the terms for the three regions as in the population case, and convergence is again to a distance of less than $c_3 s$ from a local minimizer. The probability of achieving this rate is obtained by taking a union bound over the probability of initialization in $\mathcal{C}_{\zeta_0}$ (given in Lemma 3) and the probabilities of the bounds on the gradient fluctuations holding (from Lemma 10 and (38)). Note that the fluctuation bound events imply by construction the event $\mathcal{E}_\infty = \{1 \leq \|\boldsymbol{X}\|_\infty \leq 4\sqrt{\log(np)}\}$ hence we can replace $\|\boldsymbol{X}\|_\infty$ in the conditions on $\eta$ above by $4\sqrt{\log(np)}$. The conditions on $\eta, \mu$ can be satisfied by requiring $\eta < \frac{c_5 \theta s}{n \log np}, \mu < \frac{c_6 \sqrt{\zeta_0}}{n^{5/4}}$ for suitably chosen $c_5, c_6 > 0$. The bound on the number of iterations can be simplified to the form in the theorem statement as in the population case. $\qquad \square$

## C GENERALIZED PHASE RETRIEVAL

We show below that negative curvature normal to stable manifolds of saddle points in strict saddle functions is a feature that is found not only in dictionary learning, and can be used to obtain efficient convergence rates for other nonconvex problems as well, by presenting an analysis of generalized phase retrieval that is along similar lines to the dictionary learning analysis. We stress that this contribution is not novel since a more thorough analysis was carried out by (8). The resulting rates are also suboptimal, and pertain only to the population objective.

Generalized phase retrieval is the problem of recovering a vector $\boldsymbol{x} \in \mathbb{C}^n$ given a set of magnitudes of projections $y_k = |\boldsymbol{x}^* \boldsymbol{a}_k|$ onto a known set of vectors $\boldsymbol{a}_k \in \mathbb{C}^n$. It arises in numerous domains including microscopy (27), acoustics (2), and quantum mechanics (10) (see (33) for a review). Clearly $\boldsymbol{x}$ can only be recovered up to a global phase. We consider the setting where the elements of every $\boldsymbol{a}_k$ are i.i.d. complex Gaussian, (meaning $(a_k)_j = u + iv$ for $u, v \sim \mathcal{N}(0, 1/\sqrt{2})$). We analyze the least squares formulation of the problem (7) given by

$$\min_{\boldsymbol{z} \in \mathbb{C}^n} f(\boldsymbol{z}) = \frac{1}{2p} \sum_{k=1}^p \left( y_k^2 - |\boldsymbol{z}^* \boldsymbol{a}_k|^2 \right)^2.$$

Taking the expectation (large $p$ limit) of the above objective and organizing its derivatives using Wirtinger calculus (22), we obtain

$$\mathbb{E}[f] = \|\boldsymbol{x}\|^4 + \|\boldsymbol{z}\|^4 - \|\boldsymbol{x}\|^2 \|\boldsymbol{z}\|^2 - |\boldsymbol{x}^* \boldsymbol{z}|^2 \tag{22}$$

$$\nabla \mathbb{E}[f] = \begin{bmatrix} \nabla_{\boldsymbol{z}} \mathbb{E}[f] \\ \nabla_{\overline{\boldsymbol{z}}} \mathbb{E}[f] \end{bmatrix}$$

$$= \begin{bmatrix} \left( (2\|\boldsymbol{z}\|^2 - \|\boldsymbol{x}\|^2) \boldsymbol{I} - \boldsymbol{x}\boldsymbol{x}^* \right) \boldsymbol{z} \\ \left( (2\|\boldsymbol{z}\|^2 - \|\boldsymbol{x}\|^2) \boldsymbol{I} - \overline{\boldsymbol{x}}\boldsymbol{x}^T \right) \overline{\boldsymbol{z}} \end{bmatrix}.$$

For the remainder of this section, we analyze this objective, leaving the consideration of finite sample size effects to future work.

### C.1 THE GEOMETRY OF THE OBJECTIVE

In (37) it was shown that aside from the manifold of minima

$$\breve{A} \equiv \boldsymbol{x}e^{i\theta},$$

the only critical points of $\mathbb{E}[f]$ are a maximum at $\boldsymbol{z} = \boldsymbol{0}$ and a manifold of saddle points given by

$$\hat{A} \setminus \{\boldsymbol{0}\} \equiv \left\{ \boldsymbol{z} \,\middle|\, \boldsymbol{z} \in W, \|\boldsymbol{z}\| = \frac{\|\boldsymbol{x}\|}{\sqrt{2}} \right\}$$

where $W \equiv \{z | z^* x = 0\}$. We decompose $z$ as

$$z = w + \zeta e^{i\phi} \frac{x}{\|x\|}, \tag{23}$$

where $\zeta > 0, w \in W$. This gives $\|z\|^2 = \|w\|^2 + \zeta^2$. The choice of $w, \zeta, \phi$ is unique up to factors of $2\pi$ in $\phi$, as can be seen by taking an inner product with $x$. Since the gradient decomposes as follows:

$$\nabla_z \mathbb{E}[f] = \left( 2 \|z\|^2 I - \|x\|^2 I - xx^* \right) \left( w + \zeta e^{i\phi} \frac{x}{\|x\|} \right)$$

$$= \left( 2 \|z\|^2 - \|x\|^2 \right) w + 2\zeta e^{i\phi} \left( \|z\|^2 - \|x\|^2 \right) \frac{x}{\|x\|} \tag{24}$$

the directions $e^{i\phi} \frac{x}{\|x\|}, \frac{w}{\|w\|}$ are unaffected by gradient descent and thus the problem reduces to a two-dimensional one in the space $(\zeta, \|w\|)$. Note also that the objective for this two-dimensional problem is a Morse function, despite the fact that in the original space there was a manifold of saddle points. It is also clear from this decomposition of the gradient that the stable manifolds of the saddles are precisely the set $W$.

It is evident from 24 that the dispersive property does not hold globally in this case. For $z \notin B_{\|x\|}$ we see that gradient descent will cause $\zeta$ to decrease, implying positive curvature normal to the stable manifolds of the saddles. This is a consequence of the global geometry of the objective. Despite this, in the region of the space that is more "interesting", namely $B_{\|x\|}$, we do observe the dispersive property, and can use it to obtain a convergence rate for gradient descent.

We define a set that contains the regions that feeds into small gradient regions around saddle points within $B_{\|x\|}$ by

$$\overline{Q}_{\zeta_0} \equiv \{z(\zeta, \|w\|) | \zeta \leq \zeta_0\}.$$

We will show that, as in the case of orthogonal dictionary learning, we can both bound the probability of initializing in (a subset of) the complement of $\overline{Q}_{\zeta_0}$ and obtain a rate for convergence of gradient descent in the case of such an initialization. [9]

We now define four regions of the space which will be used in the analysis of gradient descent:

$$S_1 \equiv \left\{ z \ \middle| \ \|z\|^2 \leq \tfrac{1}{2} \|x\|^2 \right\}$$

$$S_2 \equiv \left\{ z \ \middle| \ \tfrac{1}{2} \|x\|^2 < \|z\|^2 \leq (1-c) \|x\|^2 \right\}$$

$$S_3 \equiv \left\{ z \ \middle| \ (1-c) \|x\|^2 < \|z\|^2 \leq \|x\|^2 \right\}$$

$$S_4 \equiv \left\{ z \ \middle| \ \|x\|^2 < \|z\|^2 \leq (1+c) \|x\|^2 \right\}$$

defined for some $c < \tfrac{1}{4}$. These are shown in Figure 4.

We now define

$$z' \equiv z - \eta \nabla_z \mathbb{E}[f] \equiv w' + \zeta' e^{i\phi} \frac{x}{\|x\|} \tag{25}$$

and using 24 obtain

$$\zeta' = \left( 1 - 2\eta(\|z\|^2 - \|x\|^2) \right) \zeta \tag{26a}$$

$$\|w'\| = \left( 1 - \eta \left( 2 \|z\|^2 - \|x\|^2 \right) \right) \|w\|. \tag{26b}$$

---

[9] $\overline{Q}_{\zeta_0}$ is equivalent to the complement of the set $C_\zeta$ used in the analysis of the separable objective and dictionary learning.

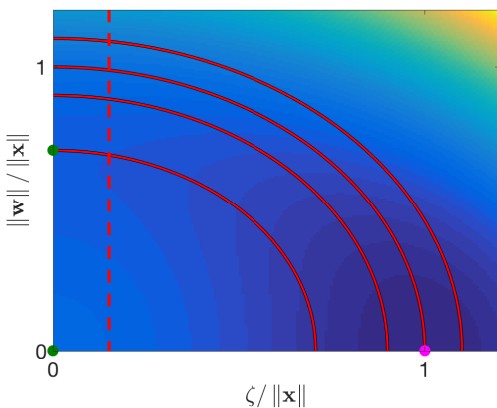

Figure 4: The projection of the objective of generalized phase retrieval on the $(\frac{\zeta}{\|\boldsymbol{x}\|}, \frac{\|\boldsymbol{w}\|}{\|\boldsymbol{x}\|})$ plane. The full red curves are the boundaries between the sets $S_1, S_2, S_3, S_4$ used in the analysis. The dashed red line is the boundary of the set $\overline{Q}_{\zeta_0}$ that contains small gradient regions around critical points that are not minima. The maximizer and saddle point are shown in dark green, while the minimizer is in pink.

These are used to find the change in $\zeta, \|\boldsymbol{w}\|$ at every iteration in each region:

$$\textbf{On } S_1\textbf{:} \qquad \zeta' \geq (1 + \eta \|\boldsymbol{x}\|^2)\zeta \tag{27a}$$

$$\|\boldsymbol{w}'\| \geq \|\boldsymbol{w}\| \tag{27b}$$

$$\textbf{On } S_2\textbf{:} \qquad \zeta' \geq (1 + 2c\eta \|\boldsymbol{x}\|^2)\zeta \tag{27c}$$

$$\|\boldsymbol{w}'\| \leq \|\boldsymbol{w}\| \tag{27d}$$

$$\textbf{On } S_3\textbf{:} \qquad \left(1 - \eta \|\boldsymbol{x}\|^2\right) \|\boldsymbol{w}\| \leq \|\boldsymbol{w}'\|$$

$$\leq \left(1 - (1 - 2c)\eta \|\boldsymbol{x}\|^2\right) \|\boldsymbol{w}\| \tag{27e}$$

$$\zeta \leq \zeta' \leq (1 + 2c\eta \|\boldsymbol{x}\|^2)\zeta \tag{27f}$$

$$\textbf{On } S_4\textbf{:} \qquad \left(1 - (1 + 2c)\eta \|\boldsymbol{x}\|^2\right) \|\boldsymbol{w}\| \leq \|\boldsymbol{w}'\|$$

$$\leq \left(1 - \eta \|\boldsymbol{x}\|^2\right) \|\boldsymbol{w}\| \tag{27g}$$

$$(1 - 2c\eta \|\boldsymbol{x}\|^2)\zeta \leq \zeta' \leq \zeta \tag{27h}$$

## C.2 Behavior of gradient descent in $\cup_{i=1}^4 S_i$

We now show that gradient descent initialized in $S_1 \backslash \overline{Q}_{\zeta_0}$ cannot exit $\cup_{i=1}^4 S_i$ or enter $\overline{Q}_{\zeta_0}$. Lemma 14 guarantees that gradient descent initialized in $\cup_{i=1}^4 S_i$ remains in this set. From equation 27 we see that a gradient descent step can only decrease $\zeta$ if $\boldsymbol{z} \in S_4$. Under the mild assumption $\zeta_0^2 < \frac{7}{16} \|\boldsymbol{x}\|^2$ we are guaranteed from Lemma 13 that at every iteration $\zeta \geq \zeta_0$. Thus the region with $\zeta < \zeta_0$ can only be entered if gradient descent is initialized in it. It follows that initialization in $S_1 \backslash \overline{Q}_{\zeta_0}$ rules out entering $\overline{Q}_{\zeta_0}$ at any future iteration of gradient descent. Since this guarantees that regions that feed into small gradient regions are avoided, an efficient convergence rate can again be obtained.

## C.3 Convergence rate

**Theorem 3** (Gradient descent convergence rate for generalized phase retrieval)**.** *Gradient descent on 22 with step size $\eta < \frac{\sqrt{c}}{4\|\boldsymbol{x}\|^2}, c < \frac{1}{4}$, initialized uniformly in $S_1$ converges to a point*

$\boldsymbol{z}$ *such that* $\mathrm{dist}(\boldsymbol{z}, \check{A}) < \sqrt{5c}\, \|\boldsymbol{x}\|$ *in*

$$T < \frac{\log\left(\frac{\|\boldsymbol{x}\|}{\zeta\sqrt{2}}\right)}{\log(1+\eta\|\boldsymbol{x}\|^2)} + \frac{\log(2)}{2\log(1+2c\eta\|\boldsymbol{x}\|^2)} + \frac{\log(2c)\log(\frac{4}{\sqrt{7}})}{\log\left(1-(1-2c)\eta\|\boldsymbol{x}\|^2\right)\log(1+2c\eta\|\boldsymbol{x}\|^2)}$$

*iterations with probability*

$$\mathbb{P} \geq 1 - \sqrt{\frac{8}{\pi}}\,\mathrm{erf}\left(\frac{\sqrt{2n}}{\|\boldsymbol{x}\|}\zeta\right),$$

*Proof.* Please see Appendix C.3. $\qquad\square$

We find that in order to prevent the failure probability from approaching 1 in a high dimensional setting, if we assume that $\|\boldsymbol{x}\|$ does not depend on $n$ we require that $\zeta$ scale like $\frac{1}{\sqrt{n}}$. This is simply the consequence of the well-known concentration of volume of a hypersphere around the equator. Even with this dependence the convergence rate itself depends only logarithmically on dimension, and this again is a consequence of the logarithmic dependence of $\zeta$ due to the curvature properties of the objective.

**Lemma 13.** *For any iterate $\boldsymbol{z}$ of gradient descent on 22, assuming $\eta < \frac{\sqrt{c}}{4\|\boldsymbol{x}\|^2}, c < \frac{1}{4}$ and defining $\zeta'$ as in 25, we have i)*

$$\boldsymbol{z} \in \bigcup_{i=1}^{4} S_i \Rightarrow \|\boldsymbol{w}\|^2 \leq \frac{\|\boldsymbol{x}\|^2}{2}$$

*ii)*

$$\boldsymbol{z} \in S_4 \Rightarrow \zeta'^2 \geq \frac{7}{16}\|\boldsymbol{x}\|^2$$

**Proof of Lemma 13.** i) From 27 we see that in $\bigcup\limits_{i=2}^{4} S_i$ the quantity $\|\boldsymbol{w}\|^2$ cannot increase, hence this can only happen in $S_1$. We show that for some $\boldsymbol{z} \in S_1$, a point with $\|\boldsymbol{w}\| = (1-\varepsilon)\frac{\|\boldsymbol{x}\|}{\sqrt{2}}, \varepsilon < 1$ cannot reach a point with $\|\boldsymbol{w}\|' = \frac{\|\boldsymbol{x}\|}{\sqrt{2}}$ by a gradient descent step. This would mean

$$\left(1 - \eta\left(2\|\boldsymbol{w}\|^2 + 2\zeta^2 - \|\boldsymbol{x}\|^2\right)\right)\|\boldsymbol{w}\|$$
$$= \left(1 - \eta\left((1-\varepsilon)^2\|\boldsymbol{x}\|^2 + 2\zeta^2 - \|\boldsymbol{x}\|^2\right)\right)(1-\varepsilon)\frac{\|\boldsymbol{x}\|}{\sqrt{2}}$$
$$= \frac{\|\boldsymbol{x}\|}{\sqrt{2}}$$

and since $\zeta^2 \geq 0$ this implies

$$\left(1 + \varepsilon\eta\|\boldsymbol{x}\|^2(2-\varepsilon)\right)(1-\varepsilon) \geq 1$$

by considering the product of these two factors, this in turn implies

$$\frac{1}{2b}(2-\varepsilon) \geq \eta\|\boldsymbol{x}\|^2(2-\varepsilon) \geq 1$$

where we have used $\eta < \frac{\sqrt{c}}{b\|\boldsymbol{x}\|^2}, c < \frac{1}{4}$. Thus if we choose $b = 4$ this inequality cannot be satisfied.

Additionally, if we initialize in $S_1 \cap \overline{Q}_{\zeta_0}$ then we cannot initialize at a point where $\|\boldsymbol{w}\|' = \frac{\|\boldsymbol{x}\|}{\sqrt{2}}$ and hence the inequality is strict.

ii) Since only a step from $S_4$ can decrease $\zeta$, we have that for the initial point $\|z\|^2 > \|x\|^2$. Combined with $\|w\|^2 \leq \frac{\|x\|^2}{2}$ this gives

$$\zeta^2 \geq \frac{\|x\|^2}{2}$$

and using the lower bound $(1 - 2\eta \|x\|^2 c)\zeta \leq \zeta'$ we obtain

$$\zeta'^2 \geq \frac{\|x\|^2}{2}(1 - 2\eta \|x\|^2 c)^2 \geq \frac{\|x\|^2}{2}(1 - 4\eta \|x\|^2 c)$$

$$\geq (1 - \frac{1}{2b})\frac{\|x\|^2}{2}$$

where in the last inequality we used $c < \frac{1}{4}, \eta < \frac{\sqrt{c}}{b\|x\|^2}$. Choosing $b = 4$ gives

$$\zeta'^2 \geq \frac{7}{16} \|x\|^2$$

If we require $\zeta_0^2 < \frac{7}{16} \|x\|^2$ this also ensures that the next iterate cannot lie in the small gradient regions around the stable manifolds of the saddles. □

**Lemma 14.** *Defining $z'$ as in 25, under the conditions of Lemma 13 and we have*

*i)*

$$z \in \bigcup_{i=2}^{4} S_i \Rightarrow z' \in \bigcup_{i=2}^{4} S_i$$

*ii)*

$$z \in S_1 \Rightarrow z' \in S_1 \cup S_2$$

**Proof of Lemma 14.** We use the fact that for the next iterate we have

$$\|z'\|^2 = \left(1 - \eta(2\|z\|^2 - \|x\|^2)\right)^2 \|w\|^2 \tag{28}$$
$$+ \left(1 - 2\eta(\|z\|^2 - \|x\|^2)\right)^2 \zeta^2$$

We will also repeatedly use $\eta < \frac{\sqrt{c}}{b\|x\|^2}, c < \frac{1}{4}$ and $z \in \bigcup_{i=1}^{4} S_i \Rightarrow \|w\|^2 \leq \frac{\|x\|^2}{2}$ which is a shown in Lemma 13.

## C.4  $z \in S_3 \Rightarrow z' \in \bigcup_{i=2}^{4} S_i$

We want to show $\frac{\|x\|^2}{2} \underset{(1)}{<} \|z'\|^2 \underset{(2)}{\leq} (1 + c)\|x\|^2$.

1) We have $z \in S_3 \Rightarrow \|z\|^2 = (1 - \varepsilon)\|x\|^2$ for some $\varepsilon \leq c$ and using 28 we must show

$$\frac{\|x\|^2}{2} \leq \begin{array}{l} \left(1 - \eta \|x\|^2 (1 - 2\varepsilon)\right)^2 \|w\|^2 \\ + \left(1 + 2\eta \|x\|^2 \varepsilon\right)^2 \zeta^2 \end{array}$$

or equivalently

$$A \equiv \varepsilon - \frac{\|x\|^2}{2}$$

$$\leq \eta \left\| \boldsymbol{x} \right\|^2 \left[ \begin{array}{c} \left( -2(1-2\varepsilon) + (1-2\varepsilon)^2 \eta \left\| \boldsymbol{x} \right\|^2 \right) \left\| \boldsymbol{w} \right\|^2 \\ +4 \left( \varepsilon + \varepsilon^2 \eta \left\| \boldsymbol{x} \right\|^2 \right) \zeta^2 \end{array} \right] \equiv B$$

and using $\eta < \frac{\sqrt{c}}{b \| \boldsymbol{x} \|^2}, c < \frac{1}{4}$

$$\frac{-\left\| \boldsymbol{x} \right\|^2}{b} < \frac{-2 \left\| \boldsymbol{x} \right\| \sqrt{c}}{b} < -2\eta \left\| \boldsymbol{x} \right\|^4 \leq B$$

while on the other hand

$$A \leq c - \frac{\left\| \boldsymbol{x} \right\|^2}{2} < -\frac{\left\| \boldsymbol{x} \right\|^2}{4}$$

thus picking $b = 4$ guarantees the desired result.

2) By a similar argument, $\left\| \boldsymbol{z}' \right\|^2 \leq (1+c) \left\| \boldsymbol{x} \right\|^2$ is equivalent to

$$A \equiv \eta \left\| \boldsymbol{x} \right\|^2 \left[ \begin{array}{c} \left( -2(1-2\varepsilon) + \eta \left\| \boldsymbol{x} \right\|^2 (1-2\varepsilon)^2 \right) \left\| \boldsymbol{w} \right\|^2 \\ +4 \left( \varepsilon + \eta \left\| \boldsymbol{x} \right\|^2 \varepsilon^2 \right) \zeta^2 \end{array} \right]$$

$$\leq \left\| \boldsymbol{x} \right\|^2 (c + \varepsilon) \equiv B$$

. Since $\left\| \boldsymbol{w} \right\|^2 \leq \frac{\| \boldsymbol{x} \|^2}{2}$ and $\left\| \boldsymbol{z} \right\|^2 \leq \left\| \boldsymbol{x} \right\|^2 \Rightarrow \zeta^2 \leq \frac{\| \boldsymbol{x} \|^2}{2}$ we obtain

$$A \leq \eta \left[ \eta \left\| \boldsymbol{x} \right\|^4 + 4 \left( \left\| \boldsymbol{x} \right\|^2 \varepsilon + \eta \left\| \boldsymbol{x} \right\|^4 \varepsilon^2 \right) \right] \frac{\left\| \boldsymbol{x} \right\|^2}{2}$$

$$< \frac{1}{2b} \left[ \frac{1}{b} + 2 \left( 1 + \frac{1}{8b} \right) \right] c \left\| \boldsymbol{x} \right\|^2$$

. If we choose $b = 4$ we thus have $A < B$ which implies

$$\left\| \boldsymbol{z}' \right\|^2 < (1+c) \left\| \boldsymbol{x} \right\|^2$$

C.5  $\boldsymbol{z} \in S_4 \Rightarrow \boldsymbol{z}' \in \bigcup\limits_{i=2}^{4} S_i$

We have $\boldsymbol{z} \in S_4 \Rightarrow \left\| \boldsymbol{z} \right\|^2 = \left\| \boldsymbol{w} \right\|^2 + \zeta^2 = (1+\varepsilon) \left\| \boldsymbol{x} \right\|^2$ for some $\varepsilon \leq c$ .

1) $\frac{\| \boldsymbol{x} \|^2}{2} < \left\| \boldsymbol{z}' \right\|^2$ is equivalent to

$$A \equiv -(\varepsilon + \frac{1}{2}) \left\| \boldsymbol{x} \right\|^2$$

$$\leq \eta \left\| \boldsymbol{x} \right\|^2 \left[ \begin{array}{c} \left( -4(1+2\varepsilon) + \eta \left\| \boldsymbol{x} \right\|^2 (1+2\varepsilon)^2 \right) \left\| \boldsymbol{w} \right\|^2 \\ +4 \left( -\varepsilon + \eta \left\| \boldsymbol{x} \right\|^2 \varepsilon^2 \right) \zeta^2 \end{array} \right] \equiv B$$

. We have

$$B \geq -4\eta \left\| \boldsymbol{x} \right\|^2 \left( (1+2\varepsilon) \left\| \boldsymbol{w} \right\|^2 + \varepsilon \zeta^2 \right) \geq -\frac{15}{8b} \left\| \boldsymbol{x} \right\|^2$$

where the last inequality used $\left\| \boldsymbol{w} \right\|^2 \leq \frac{\| \boldsymbol{x} \|^2}{2}$ and $\left\| \boldsymbol{z} \right\|^2 \leq \left\| \boldsymbol{x} \right\|^2 (1+c) \Rightarrow \zeta^2 \leq \left\| \boldsymbol{x} \right\|^2 (\frac{1}{2} + c)$. The choice $b = 4$ gaurantees $A \leq B$ which ensures the desired result.

2) This is trivial since $\left\| \boldsymbol{z} \right\|^2 \leq (1+c) \left\| \boldsymbol{x} \right\|^2$ and in $S_4$ both $\zeta$ and $\left\| \boldsymbol{w} \right\|$ decay at every iteration (ref eq).

## C.6  $z \in S_2 \Rightarrow z' \in \bigcup_{i=2}^{4} S_i$

1) We use $z \in S_2 \Rightarrow \|z\|^2 = \|w\|^2 + \zeta^2 = (\frac{1}{2} + \varepsilon)\|x\|^2$ for some $\varepsilon \leq \frac{1}{2} - c$ . Using a similar argument as in the previous section, we are required to show

$$-\varepsilon\|x\|^2 < \eta\|x\|^2 \left[ \begin{array}{c} 4\left(-\varepsilon + \varepsilon^2\eta\|x\|^2\right)\|w\|^2 \\ + \left(2(1-2\varepsilon) + (1-2\varepsilon)^2\eta\|x\|^2\right)\zeta^2 \end{array} \right]$$
$$\equiv B$$

where $B \geq -\varepsilon\frac{\|x\|^2}{b}$ implies that $b = 4$ gives the desired result.

2) The condition is equivalent to

$$A \equiv \eta\|x\|^2 \left[ \begin{array}{c} 4\left(-\varepsilon + \varepsilon^2\eta\|x\|^2\right)\|w\|^2 \\ + \left(2(1-2\varepsilon) + (1-2\varepsilon)^2\eta\|x\|^2\right)\zeta^2 \end{array} \right] + \varepsilon\|x\|^2$$
$$\leq (\frac{1}{2} + c)\|x\|^2 \equiv B$$

One can show by looking for critical points of $A(\varepsilon)$ in the range $0 \leq \varepsilon \leq \frac{1}{2}$ that $A$ is maximized at $\varepsilon = 0$, since there is only one critical point at $\varepsilon^* = \frac{4 - \frac{b}{\sqrt{c}} + 2\frac{\sqrt{c}}{b}}{8\frac{\sqrt{c}}{b}}$ and $A(\varepsilon^*) < 0$, while

$$A(\frac{1}{2}) \leq \left[ \left(-2\frac{\sqrt{c}}{b} + \frac{c}{b^2}\right)\|w\|^2 \right] + \frac{1}{2}\|x\|^2$$

$$A(0) \leq \frac{1}{2b}\left(2 + \frac{1}{2b}\right)\frac{\|x\|^2}{2}$$

and in both cases $b = 4$ ensures $A \leq B$.

## C.7  $z \in S_1 \Rightarrow z' \in S_1 \cup S_2$

We must show $\|z'\| \leq (1-c)\|x\|^2$ using $\|z\|^2 = (1-\varepsilon)\frac{\|x\|^2}{2}$ for $0 \leq \varepsilon \leq 1$.

$$\|z'\|^2 = \left(1 + \varepsilon\eta\|x\|^2\right)^2\|w\|^2 + \left(1 + 2(\varepsilon+1)\eta\frac{\|x\|^2}{2}\right)^2\zeta^2$$

$$A \equiv \eta\|x\|^2 \left[ \begin{array}{c} \left(2\varepsilon + \varepsilon^2\eta\|x\|^2\right)\|w\|^2 \\ + \left(2(\varepsilon+1) + (\varepsilon+1)^2\eta\frac{\|x\|^2}{4}\right)\zeta^2 \end{array} \right] - \varepsilon\|x\|^2$$
$$\leq (\frac{1}{2} - c)\|x\|^2 \equiv B$$

and since $A \leq \frac{1}{2b}\left[2 + \frac{1}{b}\right]\frac{\|x\|^2}{2}$ and $B \geq \frac{\|x\|^2}{4}$ once again $b = 4$ suffices to obtain the desired result.  □

**Lemma 15.** *For $z$ parametrized as in 23,*
$$\|w\|^2 < c\|x\|^2 \vee \zeta^2 > (1-c)\|x\|^2$$
$$\Rightarrow \mathrm{dist}(z, \check{A}) < \sqrt{5c}\|x\|$$

**Proof of Lemma 15.** Once $\|\boldsymbol{w}\|^2 < c \|\boldsymbol{x}\|^2$ for some $\boldsymbol{z} \in S_3 \cup S_4$ we have

$$\|\boldsymbol{z}\|^2 = \zeta^2 + \|\boldsymbol{w}\|^2 \geq (1 - c) \|\boldsymbol{x}\|^2$$

$$\zeta^2 \geq (1 - c) \|\boldsymbol{x}\|^2 - \|\boldsymbol{w}\|^2 > (1 - 2c) \|\boldsymbol{x}\|^2 \tag{29}$$

For some $\boldsymbol{z} = \boldsymbol{w} + \zeta e^{i\phi} \frac{\boldsymbol{x}}{\|\boldsymbol{x}\|}$ we have

$$\text{dist}^2(\boldsymbol{z}, \breve{A}) = \min_\theta \left\| e^{i\theta} \boldsymbol{x} - \boldsymbol{w} - \zeta e^{i\phi} \frac{\boldsymbol{x}}{\|\boldsymbol{x}\|} \right\|^2$$
$$= \|\boldsymbol{w}\|^2 + \min_\theta \left\| e^{i\theta} \boldsymbol{x} - \zeta e^{i\phi} \frac{\boldsymbol{x}}{\|\boldsymbol{x}\|} \right\|^2$$

$$= \|\boldsymbol{w}\|^2 + (1 - \frac{\zeta}{\|\boldsymbol{x}\|})^2 \|\boldsymbol{x}\|^2 = \|\boldsymbol{z}\|^2 + \|\boldsymbol{x}\|^2 - 2\zeta \|\boldsymbol{x}\|$$

if we assume $\|\boldsymbol{z}\|^2 \leq (1 + c) \|\boldsymbol{x}\|^2$

$$\text{dist}^2(\boldsymbol{z}, \breve{A}) \leq (c + 2) \|\boldsymbol{x}\|^2 - 2\zeta \|\boldsymbol{x}\| \tag{30}$$

plugging in the value of $\zeta$ from 29 and using fact that $-\sqrt{1 - x} \leq -1 + x$ for $x < 1$ we have

$$\text{dist}^2(\boldsymbol{z}, \breve{A}) < (c + 2) \|\boldsymbol{x}\|^2 - 2\sqrt{1 - 2c} \|\boldsymbol{x}\|^2 \leq 5c \|\boldsymbol{x}\|^2$$

Alternatively, if $\zeta^2 > (1 - c) \|\boldsymbol{x}\|^2$ we have from 30

$$\text{dist}^2(\boldsymbol{z}, \breve{A}) \leq (c + 2) \|\boldsymbol{x}\|^2 - 2\zeta \|\boldsymbol{x}\|$$
$$< (c + 2) \|\boldsymbol{x}\|^2 - 2\sqrt{1 - c} \|\boldsymbol{x}\|^2 \leq 3c \|\boldsymbol{x}\|^2$$

which gives the desired result. In particular, if we choose $c = \frac{1}{35}$ we converge to $\text{dist}^2(\boldsymbol{z}, \breve{A}) < \frac{\|\boldsymbol{x}\|^2}{7}$, a region which is strongly convex according to (38). $\qquad\square$

**Proof of Theorem 3: (Gradient descent convergence rate for generalized phase retrieval)**. We now bound the number of iterations that gradient descent, after random initialization in $S_1$, requires to reach a point where one of the convergence criteria detailed in Lemma 15 is fulfilled. From Lemma 14, we know that after initialization in $S_1$ we need to consider only the set $\bigcup_{i=1}^4 S_i$. The number of iterations in each set will be determined by the bounds on the change in $\zeta, \|\boldsymbol{w}\|$ detailed in 27.

C.7.1 ITERATIONS IN $S_1$

Assuming we initialize with some $\zeta = \zeta_0$. Then the maximal number of iterations in this region is

$$\zeta_0 (1 + \eta \|\boldsymbol{x}\|^2)^{t_1} = \frac{\|\boldsymbol{x}\|}{\sqrt{2}}$$

$$t_1 = \frac{\log \left( \frac{\|\boldsymbol{x}\|}{\zeta_0 \sqrt{2}} \right)}{\log(1 + \eta \|\boldsymbol{x}\|^2)}$$

since after this many iterations $\|\boldsymbol{z}\|^2 \geq \zeta^2 \geq \frac{\|\boldsymbol{x}\|^2}{2}$.

### C.7.2 Iterations in $\bigcup_{i=2}^{4} S_i$

The convergence criteria are $\|\boldsymbol{w}\|^2 < c\|\boldsymbol{x}\|^2$ or $\zeta^2 > (1-c)\|\boldsymbol{x}\|^2$.

After exiting $S_1$ and assuming the next iteration is in $S_2$, the maximal number of iterations required to reach $S_3 \cup S_4$ is obtained using

$$\zeta' \geq (1 + 2\eta\|\boldsymbol{x}\|^2 c)\zeta$$

and is given by

$$\frac{\|\boldsymbol{x}\|}{\sqrt{2}}(1 + 2\eta\|\boldsymbol{x}\|^2 c)^{t_2} = (1-c)\|\boldsymbol{x}\|^2$$

$$t_2 = \frac{\log\left(\sqrt{2(1-c)}\right)}{\log(1 + 2\eta\|\boldsymbol{x}\|^2 c)} \leq \frac{\log(2)}{2\log(1 + 2\eta\|\boldsymbol{x}\|^2 c)}$$

since after this many iterations $\|\boldsymbol{z}\|^2 \geq \zeta^2 \geq (1-c)\|\boldsymbol{x}\|^2$.

For every iteration in $S_3 \cup S_4$ we are guaranteed

$$\|\boldsymbol{w}'\| \leq \left(1 - (1 - 2c)\eta\|\boldsymbol{x}\|^2\right)\|\boldsymbol{w}\|$$

thus using Lemmas 13.i and 15 the number of iterations in $S_3 \cup S_4$ required for convergence is given by

$$\frac{\|\boldsymbol{x}\|^2}{2}\left(1 - (1 - 2c)\eta\|\boldsymbol{x}\|^2\right)^{t_{3+4}} = c\|\boldsymbol{x}\|^2$$

$$t_{3+4} = \frac{\log(2c)}{\log\left(1 - (1 - 2c)\eta\|\boldsymbol{x}\|^2\right)}$$

The only concern is that after an iteration in $S_3 \cup S_4$ the next iteration might be in $S_2$. To account for this situation, we find the maximal number of iterations required to reach $S_3 \cup S_4$ again. This is obtained from the bound on $\zeta$ in Lemma 13.

Using this result, and the fact that for every iteration in $S_2$ we are guaranteed $\zeta' \geq (1 + 2\eta\|\boldsymbol{x}\|^2 c)\zeta$ the number of iterations required to reach $S_3 \cup S_4$ again is given by

$$\frac{\sqrt{7}}{4}\|\boldsymbol{x}\|(1 + 2\eta\|\boldsymbol{x}\|^2 c)^{t_r} = \sqrt{1-c}\|\boldsymbol{x}\|$$

$$t_r = \frac{\log\left(\frac{4\sqrt{1-c}}{\sqrt{7}}\right)}{\log(1 + 2\eta\|\boldsymbol{x}\|^2 c)} \leq \frac{\log(\frac{4}{\sqrt{7}})}{\log(1 + 2\eta\|\boldsymbol{x}\|^2 c)}$$

### C.8 Final rate

The final rate to convergence is

$$T < t_1 + t_2 + t_{3+4}t_r$$

$$= \frac{\log\left(\frac{\|\boldsymbol{x}\|}{\zeta\sqrt{2}}\right)}{\log(1 + \eta\|\boldsymbol{x}\|^2)} + \frac{\log(2)}{2\log(1 + 2c\eta\|\boldsymbol{x}\|^2)}$$

$$+ \frac{\log(2c)\log(\frac{4}{\sqrt{7}})}{\log\left(1 - (1 - 2c)\eta\|\boldsymbol{x}\|^2\right)\log(1 + 2c\eta\|\boldsymbol{x}\|^2)}$$

### C.9 Probability of the bound holding

The bound applies to an initialization with $\zeta \geq \zeta_0$, hence in $S_1 \backslash \overline{Q}_{\zeta_0}$. Assuming uniform initialization in $S_1$, the set $\overline{Q}_{\zeta_0}$ is simply a band of width $2\zeta_0$ around the equator of the

ball $B_{\|\boldsymbol{x}\|/\sqrt{2}}$ (in $\mathbb{R}^{2n}$, using the natural identification of $\mathbb{C}^n$ with $\mathbb{R}^{2n}$). This volume can be calculated by integrating over $2n-1$ dimensional balls of varying radius.

Denoting $r = \frac{\zeta_0\sqrt{2}}{\|\boldsymbol{x}\|}$ and by $V(n) = \frac{\pi^{n/2}}{\frac{n}{2}\Gamma(\frac{n}{2})}$ the hypersphere volume, the probability of initializing in $S_1 \cap \overline{Q}_{\zeta_0}$ (and thus in a region that feeds into small gradient regions around saddle points) is

$$
\begin{aligned}
\mathbb{P}(\text{fail}) &= \frac{\text{Vol}(\overline{Q}_{\zeta_0})}{\text{Vol}(B_{\|\boldsymbol{x}\|/\sqrt{2}})} \\
&= \frac{V(2n-1)\int\limits_{-r}^{r}(1-x^2)^{\frac{2n-1}{2}}dx}{V(2n)} \\
&\leq \frac{V(2n-1)\int\limits_{-r}^{r}e^{-\frac{2n-1}{2}x^2}dx}{V(2n)} \\
&= \frac{1}{\sqrt{n-\frac{1}{2}}}\frac{n}{n-\frac{1}{2}}\frac{\Gamma(n)}{\Gamma(\frac{2n-1}{2})}\text{erf}(\sqrt{\frac{2n-1}{2}}r) \\
&\leq \sqrt{\frac{8}{\pi}}\text{erf}(\sqrt{n}r)
\end{aligned}
$$

. For small $\zeta$ we again find that $\mathbb{P}(\text{fail})$ scales linearly with $\zeta$, as was the case for the previous problems considered. $\qquad\square$

## D   Auxiliary Lemmas

### D.1   Separable objective

$$
\frac{\partial g_s(\boldsymbol{w})}{\partial w_i} = \tanh\left(\frac{w_i}{\mu}\right) - \tanh\left(\frac{q_n}{\mu}\right)\frac{w_i}{q_n}
$$

$$
\frac{\partial^2 g_s(\boldsymbol{w})}{\partial w_i \partial w_j} = \left[\frac{1}{\mu}\text{sech}^2\left(\frac{w_i}{\mu}\right) - \tanh\left(\frac{q_n}{\mu}\right)\frac{1}{q_n}\right]\delta_{ij}
$$

$$
+ \left[\frac{1}{\mu}\text{sech}^2\left(\frac{q_n}{\mu}\right)\frac{1}{q_n^2} - \tanh\left(\frac{q_n}{\mu}\right)\frac{1}{q_n^3}\right]w_i w_j
$$

### D.2   Dictionary Learning

$$
\nabla_{\boldsymbol{w}} g_{DL}^{pop}(\boldsymbol{w}) = \mathbb{E}\left[\tanh\left(\frac{\boldsymbol{q}^*(\boldsymbol{w})\boldsymbol{x}}{\mu}\right)\left(\overline{x} - \frac{x_n}{q_n(\boldsymbol{w})}w\right)\right]
$$

### D.3   Properties of $\mathcal{C}_\zeta$

**Proof of Lemma 3: (Volume of $\mathcal{C}_\zeta$).** We are interested in the relative volume $\frac{\text{Vol}(\mathcal{C}_\zeta)}{\text{Vol}(\mathbb{S}^{n-1})} \equiv V_\zeta$. Using the standard solid angle formula, it is given by

$$
V_\zeta = \lim_{\varepsilon \to 0}\frac{1}{\varepsilon^{n/2}}\int\limits_{0}^{\infty}e^{-\frac{\pi}{\varepsilon}x_1^2}\prod_{i=2}^{n}\int\limits_{-x_1/(1+\zeta)}^{x_1/(1+\zeta)}e^{-\frac{\pi}{\varepsilon}x_i^2}dx_i dx_1
$$

$$
= \lim_{\varepsilon \to 0}\frac{1}{\sqrt{\varepsilon}}\int\limits_{0}^{\infty}e^{-\frac{\pi}{\varepsilon}x^2}\left[\text{erf}(\frac{x}{(1+\zeta)}\sqrt{\frac{\pi}{\varepsilon}})\right]^{n-1}dx
$$

changing variables to $\tilde{x} = \sqrt{\frac{\pi}{\varepsilon}} \frac{x}{(1+\zeta)}$

$$V_\zeta = \frac{(1+\zeta)}{\sqrt{\pi}} \int\limits_0^\infty e^{-(1+\zeta)^2 x^2} \mathrm{erf}^{n-1}(x) dx$$

This integral admits no closed form solution but one can construct a linear approximation around small $\zeta$ and show that it is convex. Thus the approximation provides a lower bound for $V_\zeta$ and an upper bound on the failure probability.

From symmetry considerations the zero-order term is $V_0 = \frac{1}{2n}$. The first-order term is given by

$$\frac{\partial V_\zeta}{\partial \zeta}\bigg|_{\zeta=0} = \frac{1}{n} - \frac{2}{\sqrt{\pi}} \int\limits_0^\infty x^2 e^{-x^2} \mathrm{erf}^{n-1}(x) dx$$

We now require an upper bound for the second integral since we are interested in a lower bound for $V_\zeta$. We can express it in terms of the second moment of the $L^\infty$ norm of a Gaussian vector as follows:

$$\frac{1}{\sqrt{\pi}} \int\limits_0^\infty x^2 e^{-x^2} \mathrm{erf}^{n-1}(x) = \frac{1}{\sqrt{\pi}} \int\limits_0^\infty x^2 e^{-x^2} \Pi_i \frac{1}{\sqrt{\pi}} \int\limits_{-x}^x e^{-t_i^2} dt_i dx$$

$$= \frac{1}{\sqrt{2\pi}} \int\limits_0^\infty \frac{x^2}{2} e^{-x^2/2} \Pi_i \frac{1}{\sqrt{2\pi}} \int\limits_{-x}^x e^{-t_i^2/2} dt_i dx$$

$$= \frac{1}{4n} \int \|\boldsymbol{X}\|_\infty^2 \, d\mu(\boldsymbol{X})$$

$$= \frac{1}{4n} \left( \mathrm{Var}\left[\|\boldsymbol{X}\|_\infty\right] + \left(\mathbb{E}\left[\|\boldsymbol{X}\|_\infty\right]\right)^2 \right)$$

where $\mu(\boldsymbol{X})$ is the Gaussian measure on the vector $\boldsymbol{X} \in \mathbb{R}^n$. We can bound the first term using

$$\mathrm{Var}\left[\|\boldsymbol{X}\|_\infty\right] \leq \max_i \mathrm{Var}\left[|X_i|\right] = \mathrm{Var}\left[|X_i|\right] < \mathrm{Var}\left[X_i\right] = 1$$

To bound the second term, we use the fact that for a standard Gaussian vector $\boldsymbol{X}$ ($X_i \sim \mathcal{N}(0,1)$) and any $\lambda > 0$ we have

$$\exp\left(\lambda \mathbb{E}\left[\|\boldsymbol{X}\|_\infty\right]\right) \leq \mathbb{E}\left[\exp\left(\lambda \max_i |X_i|\right)\right]$$

$$\leq \mathbb{E}\left[\sum_i \exp\left(\lambda |X_i|\right)\right] = n\mathbb{E}\left[\exp\left(\lambda |X_i|\right)\right]$$

(using convexity and non-negativity of the exponent respectively)

$$n\mathbb{E}\left[\exp\left(\lambda |X_i|\right)\right] = 2n \int\limits_0^\infty \exp\left(\lambda X_i\right) d\mu(X_i)$$

$$\leq 2n\mathbb{E}\left[\exp\left(\lambda X_i\right)\right] = 2n \exp\left(\frac{\lambda^2}{2}\right)$$

taking the log of both sides gives

$$\mathbb{E}\left[\max_i |X_i|\right] \le \frac{\log(2n)}{\lambda} + \frac{\lambda}{2}$$

and the bound is minimized for $\lambda = \sqrt{2\log(2n)}$ giving

$$\mathbb{E}\left[\max_i |X_i|\right] \le \sqrt{2\log(2n)} \sim \sqrt{2\log(n)}$$

Combining these bounds, the leading order behavior of the gradient is

$$\frac{\partial V_\zeta}{\partial \zeta}\bigg|_{\zeta=0} \ge \frac{3 - 4\log(2n)}{4n} \ge -\frac{\log(n)}{n}.$$

This linear approximation is indeed a lower bound, since using integration by parts twice we have

$$\frac{\partial^2 V_\zeta}{\partial \zeta^2} = \frac{1}{\sqrt{\pi}} \int_0^\infty e^{-(1+\zeta)^2 x^2} \left( \begin{array}{c} -6(1+\zeta)x^2 \\ +4(1+\zeta)^3 x^4 \end{array} \right) \mathrm{erf}^{n-1}(x)dx$$

$$= -\frac{2(n-1)}{\pi} \int_0^\infty e^{-(1+\zeta)^2 x^2} \left(1 - 2(1+\zeta)^2 x^2\right) e^{-x^2} \mathrm{erf}^{n-2}(x)dx$$

$$= \frac{4(n-1)(n-2)(1+\zeta)}{\pi^{3/2}} \int_0^\infty e^{-((1+\zeta)^2+2)x^2} \mathrm{erf}^{n-3}(x)dx > 0$$

where the last inequality holds for any $n > 2$ since the integrand is non-negative everywhere. This gives

$$V_\zeta \ge \frac{1}{2n} - \frac{\log(n)}{n}\zeta$$

$$\qquad\qquad\qquad\qquad\qquad\qquad\qquad\qquad\qquad\qquad\qquad\qquad\qquad\qquad \square$$

**Lemma 16.** $B^\infty_{s(\zeta)}(0) \subseteq \mathcal{C}_\zeta \subseteq B^2_{\sqrt{n-1}s(\zeta)}(0)$ where $s(\zeta) = \frac{1}{\sqrt{(2+\zeta)\zeta+n}}$. $B^\infty_{s(\zeta)}(0)$ is the largest $L^\infty$ ball contained in $\mathcal{C}_\zeta$, and $B^2_{\sqrt{n-1}s(\zeta)}(0)$ is the smallest $L^2$ ball containing $\mathcal{C}_\zeta$ (where these balls are defined in terms of the $\boldsymbol{w}$ vector). All three intersect only at the points where all the coordinates of $\boldsymbol{w}$ have equal magnitude. Additionally, $\mathcal{C}_\zeta \subseteq B^\infty_{1/\sqrt{2+\zeta}}(0)$ and this is the smallest $L^\infty$ ball containing $\mathcal{C}_\zeta$.

*Proof.* Given the surface of some $L^\infty$ ball for $\boldsymbol{w}$ , we can ask what is the minimal $\zeta$ such that $\partial \mathcal{C}_{\zeta_m}$ intersects this surface. This amounts to finding the minimal $q_n$ given some $\|\boldsymbol{w}\|_\infty$. Yet this is clearly obtained by setting all the coordinates of $w$ to be equal to $\|\boldsymbol{w}\|_\infty$ (this is possible since we are guaranteed $q_n \ge \|\boldsymbol{w}\|_\infty \Rightarrow \|\boldsymbol{w}\|_\infty \le \frac{1}{\sqrt{n}}$), giving

$$\frac{\sqrt{1 - (n-1)\|\boldsymbol{w}\|_\infty^2}}{\|\boldsymbol{w}\|_\infty} = 1 + \zeta_m$$

$$\|\boldsymbol{w}\|_\infty = \frac{1}{\sqrt{(1+\zeta_m)^2 + n - 1}}$$

thus, given some $\zeta$, the maximal $L^\infty$ ball that is contained in $\mathcal{C}_\zeta$ has radius $\frac{1}{\sqrt{(2+\zeta)\zeta+n}}$. The minimal $L^\infty$ norm containing $\mathcal{C}_\zeta$ can be shown by a similar argument to be $B^\infty_{1/\sqrt{1+(1+\zeta)^2}}(0)$, where one instead maximizes $q_n$ with some fixed $\|\boldsymbol{w}\|_\infty$.

Given some surface of an $L^2$ ball, we can ask what is the minimal $\mathcal{C}_\zeta$ such that $\mathcal{C}_\zeta \subseteq B^2_r(0)$. This is equivalent to finding the maximal $\zeta_M$ such that $\partial \mathcal{C}_{\zeta_M}$ intersects the surface of the $L^2$ ball. Since $q_n$ is fixed, maximizing $\zeta$ is equivalent to minimizing $\|\boldsymbol{w}\|_\infty$. This is done by setting $\|\boldsymbol{w}\|_\infty = \frac{\|\boldsymbol{w}\|}{\sqrt{n-1}}$, which gives

$$\frac{\sqrt{1-\|\boldsymbol{w}\|^2}}{\|\boldsymbol{w}\|}\sqrt{n-1} = 1 + \zeta_M$$

$$\sqrt{\frac{n-1}{(2+\zeta_M)\zeta_M + n}} = \|\boldsymbol{w}\|$$

The statement in the lemma follows from combining these results. $\qquad\square$

**Lemma 17** (Geometric Increase in $\zeta$). *For $\boldsymbol{w} \in C_{\zeta_0} \backslash B^\infty_b$ (where $\zeta \equiv \frac{q_n}{\|\boldsymbol{w}\|_\infty} - 1$), assume $|w_i| > r \Rightarrow \boldsymbol{u}^{(i)*}\mathrm{grad}[f](\boldsymbol{q}(\boldsymbol{w})) \geq c(\boldsymbol{w})\zeta$ where $\boldsymbol{u}^{(i)}$ is defined in 7 and $1 > b > r$. Then if $\|\mathrm{grad}[f](\boldsymbol{q}(\boldsymbol{w}))\| < M$ and we define*

$$\boldsymbol{q}' \equiv \exp_{\boldsymbol{q}}(-\eta \mathrm{grad}[f](\boldsymbol{q}))$$

*for $\eta < \frac{b-r}{3M}$, defining $\zeta'$ in an analogous way to $\zeta$ we have*

$$\zeta' \geq \zeta\left(1 + \frac{\sqrt{n}}{2}\eta c(\boldsymbol{w})\right)$$

Proof: D.3

**Proof of Lemma 17:(Geometric Increase in $\zeta$).** Denoting $g \equiv \|\mathrm{grad}[f](\boldsymbol{q})\|$, we have

$$\boldsymbol{q}' = \cos(g\eta)\boldsymbol{q} - \sin(g\eta)\frac{\mathrm{grad}[f](\boldsymbol{q})}{g}$$

hence, using Lagrange remainder terms,

$$\frac{q_n'}{w_i'} = \frac{q_n - \eta \mathrm{grad}[f](\boldsymbol{q})_n - \int_0^{g\eta}\cos(t)(g\eta - t)dt\, q_n + \int_0^{g\eta}\sin(t)(g\eta - t)dt\frac{\mathrm{grad}[f](\boldsymbol{q})_n}{g}}{w_i - \eta \mathrm{grad}[f](\boldsymbol{q})_i - \int_0^{g\eta}\cos(t)(g\eta - t)dt\, w_i + \int_0^{g\eta}\sin(t)(g\eta - t)dt\frac{\mathrm{grad}[f](\boldsymbol{q})_i}{g}}$$

. We assume $w_i > 0$, and the converse case is analogous. From convexity of $\frac{1}{1+x}$

$$\frac{q_n'}{w_i'} \geq \frac{q_n}{w_i} + \left[\frac{\eta}{w_i} - \frac{\int_0^{g\eta}\sin(t)(g\eta - t)dt}{w_i g}\right] * \left(\mathrm{grad}[f](\boldsymbol{q})_i - \frac{w_i}{q_n}\mathrm{grad}[f](\boldsymbol{q})_n\right)$$

$$= \frac{q_n}{w_i} + \frac{\sin(g\eta)}{w_i g}\left(\mathrm{grad}[f](\boldsymbol{q})_i - \frac{w_i}{q_n}\mathrm{grad}[f](\boldsymbol{q})_n\right)$$

$$= \frac{q_n}{w_i} + \frac{\sin(g\eta)}{w_i g} \boldsymbol{u}^{(i)*} \mathrm{grad}[f](\boldsymbol{q}(\boldsymbol{w}))$$

We now use $\eta < \frac{b-r}{3M} < \frac{\pi}{2M} \Rightarrow g\eta < \frac{\pi}{2} \Rightarrow \sin(g\eta) \geq \frac{g\eta}{2}$ and consider two cases. If $|w_i| > r$ we use the bound on the gradient projection in the lemma statement to obtain

$$\frac{q_n'}{w_i'} \geq \frac{q_n}{w_i} + \frac{\eta}{2w_i} c(\boldsymbol{w})\zeta \geq \frac{q_n}{w_i} + \frac{\sqrt{n}}{2} \eta c(\boldsymbol{w})\zeta$$

hence

$$\frac{q_n'}{w_i'} - 1 \geq \frac{q_n}{\|\boldsymbol{w}\|_\infty} - 1 + \frac{\sqrt{n}}{2} \eta c(\boldsymbol{w})\zeta = \zeta\left(1 + \frac{\sqrt{n}}{2} \eta c(\boldsymbol{w})\right) \tag{31}$$

If $|w_i| < r$ we rule out the possibility that $|w_i'| = \|\boldsymbol{w}'\|_\infty$ by demanding $\eta < \frac{b-r}{3M}$. Since $b(b-r) < 1$ we have $1 + \frac{1}{3}b(b-r) < \sqrt{1 + b(b-r)}$ hence the requirement on $\eta$ implies

$$\eta < \frac{\sqrt{1 + b(b-r)} - 1}{gb} = \frac{-2g + \sqrt{4g^2 + 4g^2 b(b-r)}}{2g^2 b}$$

. If we now combine this with the fact that after a Riemannian gradient step $\cos(g\eta)q_i - \sin(g\eta) \leq q_i' \leq \cos(g\eta)q_i + \sin(g\eta)$, the above condition on $\eta$ implies the inequality $(*)$, which in turn ensures that $|w_i| < r \Rightarrow |w_i'| < \|\boldsymbol{w}'\|_\infty$:

$$|w_i'| < |w_i| + \sin(g\eta) < r + g\eta \underset{(*)}{<} (1 - g^2\eta^2)b - g\eta$$

$$< \cos(g\eta) \|\boldsymbol{w}\|_\infty - \sin(g\eta) \leq \|\boldsymbol{w}'\|_\infty$$

Due to the above analysis, it is evident that any $w_i'$ such that $|w_i'| = \|\boldsymbol{w}'\|_\infty$ obeys $|w_i| > r$, from which it follows that we can use 31 to obtain

$$\frac{q_n'}{\|\boldsymbol{w}'\|_\infty} - 1 = \zeta' \geq \zeta\left(1 + \frac{\sqrt{n}}{2} \eta c(\boldsymbol{w})\right)$$

$\square$

