# OpenReview forum: "Efficient Dictionary Learning with Gradient Descent"
_ICLR.cc/2019/Conference_

### Official Review · AnonReviewer2 · 2018-10-25
**I believe I miss some thing important in this paper. This paper seems not to be self contained. I do not understand the paper very well. Therefore, I have reservations about the paper.**

**Rating:** 5
**Confidence:** 2

**Review:**

This paper analyzes the surface of the complete orthogonal dictionary learning problem, and provides converge guarantees for randomly initialized gradient descent to the neighborhood of a global optimizer. The analysis relies on the negative curvature in the directions normal to the stable manifolds of all critical points that are not global minimizer.

Exploring the surface of a function and analyzing the structure of the negative curvature normal to the stable manifolds is an interesting idea. However, I believe I miss some thing important in this paper. This paper seems not to be self contained. I do not understand the paper very well. See details below. Therefore, I have reservations about the paper.

*) The terminology "stable manifolds" is used from the first page, while its formal definition is given on page 4.
*) P3, the dictionary learning problem is not formally given. It is stated in the paper that the task is to find A and X, given Y. However, what optimization problem does the author consider? Is it \min_{A, X} \|Y - A X\|_F^2? assuming both dictionary A and sparse code X are unknown or \min_{A} \|Y - A X\|_F^2 assuming only dictionary is unknown?
*) P3, second paragraph in Section 3: what is the variable q? It is not defined before.
*) P3, third paragraph in Section 3: What is the function row()? Why does row(Y) equal row(X)?
*) P3: How does the dictionary learning problem reformulate into the problem in the third paragraph of Section 3? If I understand correctly, the task is to find A, X such that A^* Y = X since A is orthogonal. Consider the first column in A and denote it by q. Then the first column of X is approximated by q^* Y. Since X is sparse, the task is to find q so that q^* Y as sparse as possible. But how about the other columns in matrix $A$?
*) The Riemannian gradient algorithm is not stated in this paper.

---

### Official Review · AnonReviewer3 · 2018-11-03
**Iteration complexity analysis of Riemannian gradient descent for orthogonal dictionary learning with sparse factors.**

**Rating:** 4
**Confidence:** 3

**Review:**

The authors analyze the convergence performance of Riemannian gradient descent based algorithm for the dictionary learning problem with orthogonal dictionary and sparse factors. They demonstrate a polynomial time convergence from a random initialization for a smooth surrogate objective for the original non-smooth one. The problem and the analysis are of interest, but I have several questions regarding the paper as follows.

My first concern is that the analysis is on a smooth surrogate of the non-smooth sparse minimization for solving the dictionary learning problem, so it is not clear what is relationship between the global minimizer of the smooth problem to the underlying true dictionary. More specifically, how far is the global minimizer of problem (1) or (2) to the true dictionary parameter, and whether they share (approximately) the space or components regarding the sparse factors. Without clarifying this, it is not well motivated why we are interested in studying the problem considered in this paper at the beginning. Intuitively, since the recovered factors are not sparse anymore, it will impact the dictionary accordingly due to the linear mapping, which may lead to a very different set of dictionary components. Thus, explicit explanation is necessary here to avoid such degenerate case.

My second concern is the eligibility of assuming the dictionary A to be an identity matrix and extending it to the general orthogonal matrix case. The analysis uses the fact that rows of A are canonical basis, i.e., each row only has one non-zero entry. I do not see a straightforward extension by replacing A to be an orthogonal matrix as the authors claimed on page 3, since then the inner product of one row of A and one column of X is not just the corresponding entry of the column of X. It will be helpful if the authors can explain this explicitly or adjust the analysis accordingly to make this valid.

Another issue is the clarity of the paper. Some statements in the paper are not very clear. Form example, on page 3, third paragraph of Section 3, what does row(Y) = row(X_0) mean? Also, in eqn (1), y_k means k-th column of Y, and in eqn (2), q_i means i-th entry of q? Since both are bold lower case letters, clear distinction will help. Moreover, the reference use (. ) instead [ .], which can be confusing sometimes.

---

### Official Review · AnonReviewer1 · 2018-11-03
**Needs some improvement.**

**Rating:** 5
**Confidence:** 4

**Review:**

The paper presents a convergence analysis for manifold gradient descent in complete dictionary learning. I have three major concerns:

(1) The optimization problem for complete orthogonal dictionary learning in this paper is very different from overcomplete dictionary learning in practice. It is actually more similar to tensor decomposition-type problems, especially after smoothing. From this point of view, it is not as interesting as the optimization problem.

Arora et al. Simple, Efficient, and Neural Algorithms for Sparse Coding, 2015

(2) Some recent works focus on analyzing gradient descent for phase retrieval and matrix sensing. These obtained results are significantly improved and near-optimal. However, the convergence rate in this paper is very loose. Besides, the paper even does not look into the last phase of gradient descent, when there exists restricted strong convexity. Thus, only sublinear convergence rate is presented.

Chen et al. Gradient Descent with Random Initialization: Fast Global Convergence for Nonconvex Phase Retrieval, 2018

The quality of this paper could be improved, if the author could sharpen their analysis.

(3) The analysis for the manifold gradient methods is something new, but not very significant. There have already been some works on manifold gradient methods. For example, the following paper has established convergence rates to second order optimal solutions for general nonconvex function over manifold.

Boumal et al. Global rates of convergence for nonconvex optimization on manifolds. 2016.

The following paper has established the asymptotic convergence to second order optimal solutions for general nonconvex function over manifold.

Lee et al. First-order Methods Almost Always Avoid Saddle Points, 2017.

---

### Meta-Review · Area_Chair1 · 2018-12-18

**Confidence:** 5
**Recommendation:** Reject

**Metareview:**

It seems that the reviewers reached a consensus that the paper is not ready for publication in ICLR. (see more details in the reviews below. )